# Neural Network Branching for Neural Network Verification

**Jingyue Lu**
University of Oxford
jingyue.lu@spc.ox.ac.uk

**M. Pawan Kumar**
University of Oxford
pawan@robots.ox.ac.uk

## Abstract

Formal verification of neural networks is essential for their deployment in safety-critical areas. Many available formal verification methods have been shown to be instances of a unified Branch and Bound (BaB) formulation. We propose a novel framework for designing an effective branching strategy for BaB. Specifically, we learn a graph neural network (GNN) to imitate the strong branching heuristic behaviour. Our framework differs from previous methods for learning to branch in two main aspects. Firstly, our framework directly treats the neural network we want to verify as a graph input for the GNN. Secondly, we develop an intuitive forward and backward embedding update schedule. Empirically, our framework achieves roughly $50\%$ reduction in both the number of branches and the time required for verification on various convolutional networks when compared to the best available hand-designed branching strategy. In addition, we show that our GNN model enjoys both horizontal and vertical transferability. Horizontally, the model trained on easy properties performs well on properties of increased difficulty levels. Vertically, the model trained on small neural networks achieves similar performance on large neural networks. Code for all experiments is available at https://github.com/oval-group/GNN_branching.

## 1 Introduction

Despite their outstanding performances on various tasks, neural networks are found to be vulnerable to adversarial examples (Goodfellow et al., 2015; Szegedy et al., 2013). The brittleness of neural networks can have costly consequences in areas such as autonomous driving, finance and healthcare. When one requires robustness to adversarial examples, traditional model evaluation approaches, which test the trained model on a hold-out set, do not suffice. Instead, formal verification of properties such as adversarial robustness becomes necessary. For instance, to ensure self-driving cars make consistent correct decisions even when the input image is slightly perturbed, the required property to verify is that the underlying neural network outputs the same correct prediction for all points within a norm ball whose radius is determined by the maximum perturbation allowed.

Several methods have been proposed for verifying properties on neural networks (NN). Bunel et al. (2018) showed that many of the available methods can be viewed as instances of a unified BaB framework. A BaB algorithm consists of two key components: branching strategies and bounding methods. Branching strategies decide how the search space is recursively split into smaller spaces. Bounding methods compute bounds of each subspace to tighten the bounds of the final objective function over the whole search space. In this work, we focus on improving the branching strategies. By directly working with a general framework, our identified algorithmic improvements can be combined with any bounding method, leading to potential performance improvement for BaB based verification algorithms.

Branching strategies have significant impacts on the overall problem-solving process, as it directly decides the total number of steps, and consequently the total time, required to solve the problem at hand. The quality of a branching strategy is even more important when NN verification problems are considered, which generally have a very large search space. Each input dimension or each activation unit can be a potential branching option and neural networks of interest often have high dimensional inputs and thousands of hidden activation units. With such a large search space, an effective branching strategy could mean a large reduction of the total number of branches required, and consequently of the time required to solve a problem. Developing an effective strategy is thus of significant importance to the success of BaB based NN verification.

So far, to the best of our knowledge, branching rules adopted by BaB based verification methods are either random selection (Katz et al., 2017; Ehlers, 2017) or hand-designed heuristics (Wang et al.,

2018b; Bunel et al., 2018; Royo et al., 2019; Bunel et al., 2019). Random selection is generally inefficient as the distribution of the best branching decision is rarely uniform. In practice, this strategy often results in exhaustive search to make a verification decision. On the other hand, hand designed heuristics often involve a trade-off between effectiveness and computational cost. For instance, strong branching is generally one of the best performing heuristics for BaB methods in terms of the number of branches, but it is computationally prohibitive as each branching decision requires an expensive exhaustive search over all possible options. The heuristics that are currently used in practice are either inspired by the corresponding dual problem when verification is formulated as an optimization problem (Bunel et al., 2018; Royo et al., 2019) or incorporating the gradient information of the neural network (Wang et al., 2018b). These heuristics normally have better computational efficiency. However, given the complex nature of the search space, it is unlikely that any hand-designed heuristic is able to fully exploit the structure of the problem and the data distribution encountered in practice. As mentioned earlier, for large size NN verification problems, a slight reduction in the quality of the branching strategy could lead to substantial increase in the total number of branches required to solve the problem. A computationally cheap but high quality branching strategy is thus much needed.

In order to exploit the inherent structure of the problem and the data, we propose a novel machine learning framework for designing a branching strategy. Our framework is both computationally efficient and effective, giving branching decisions that are of a similar quality to that of strong branching. Specifically, we make the following contributions:

- We use a graph neural network (GNN) to exploit the structure of the neural network we want to verify. The embedding vectors of the GNN are updated by a novel schedule, which is both computationally cheap and memory efficient. In detail, we mimic the forward and backward passes of the neural network to update the embedding vectors. In addition, the proposed GNN allows a customised schedule to update embedding vectors via shared parameters. That means, once training is done, the trained GNN model is applicable to various verification properties on different neural network structures.

- We train GNNs via supervised learning. We provide ways to generate training data cheaply but inclusive enough to represent branching problems at different stages of a BaB process for various verification properties. With the ability to exploit the neural network structure and a comprehensive training data set, our GNN is easy to train and converges fast.

- Our learned GNN also enjoys transferability both horizontally and vertically. Horizontally, although trained with easy properties, the learned GNN gives similar performance on medium and difficult level properties. More importantly, vertically, given that all other parts of BaB algorithms remain the same, the GNN trained on small networks performs well on large networks. Since the network size determines the total cost for generating training data and is positively correlated with the difficulty of learning, this vertical transferability allows our framework to be readily applicable to large scale problems.

- We further enhance our framework via online learning. For a learned branching strategy, it is expected that the strategy can fail to output satisfactory branching decisions from time to time. To deal with this issue, we provide an online scheme for fine-tuning the GNN along the BaB process in order to best accommodate the verification property at hand.

- Finally, we supply a dataset on convolutional NN verification problems, covering problems at different difficulty levels over neural networks of different sizes. We hope that by providing a large problem dataset it could allow easy comparisons among existing methods and additionally encourage the development of better methods.

Since most verification methods available work on ReLU-based deep neural networks, we focus on neural networks with ReLU activation units in this paper. However, we point out that our framework is applicable to any neural network architecture.

## 2 RELATED WORKS

Learning has already been used in solving combinatorial optimization problems (Bello et al., 2016; Dai et al., 2017) and mixed integer linear programs (MILP) (Khalil et al., 2016; Alvarez et al., 2017; Hansknecht et al., 2018; Gasse et al., 2019). In these areas, instances of the same underlying structure are solved multiple times with different data values, which opens the door for learning. Among them, Khalil et al. (2016), Alvarez et al. (2017), Hansknecht et al. (2018), and Gasse et al. (2019) proposed learned branching strategies for solving MILP with BaB algorithms. These meth-

ods imitate the strong branching strategy. Specifically, Khalil et al. (2016) and Hansknecht et al. (2018) learn a ranking function to rank potential branching decisions while Alvarez et al. (2017) uses regression to assign a branching score to each potential branching choice. Apart from imitation, Anderson et al. (2019) proposed utilizing Bayesian optimization to learn verification policies. There are two main issues with these methods. Firstly, they rely heavily on hand-designed features or priors and secondly, they use a generic learning structure which is unable to exploit the neural network architecture.

The approach most relevant to ours is the concurrent work by Gasse et al. (2019). They managed to reduce feature reliance by exploiting the bipartite structure of an MILP through a GNN. The bipartite graph is capable of capturing the network architecture, but cannot exploit it effectively. Specifically, it treats all the constraints the same and updates them simultaneously using the same set of parameters. This limited expressiveness can result in a difficulty in learning and hence in a high generalization error for NN verification problems. Our proposed framework is specifically designed for NN verification problems. By exploiting the neural network structure, and designing a customized schedule for embedding updates, our framework is able to scale elegantly both in terms of computation and memory. Finally, we mention that the recently proposed verification methods (Katz et al., 2019; Singh et al., 2018; Anderson et al., 2019) are not explicitly formulated as BaBs. Since our focus is on branching, we mainly use the methods in Bunel et al. (2019) for comparison.

## 3 BACKGROUND

Formal verification of neural networks refers to the problem of proving or disproving a property over a bounded input domain. Properties are functions of neural network outputs. When a property can be expressed as a Boolean expression over linear forms, we can modify the neural network in a suitable way so that the property can be simplified to checking the sign of the neural network output (Bunel et al., 2018). Note that all the properties studied in previous works satisfy this form, thereby allowing us to use the aforementioned simplification. Mathematically, given the modified neural network $f$, a bounded input domain $\mathcal{C}$, formal verification examines the truthfulness of the following statement:

$$\forall \boldsymbol{x} \in \mathcal{C}, \qquad f(\boldsymbol{x}) \geq 0. \tag{1}$$

If the above statement is true, the property holds. Otherwise, the property does not hold.

### 3.1 BRANCH AND BOUND

Verification tasks are often treated as a global optimization problem. We want to find the minimum of $f(\boldsymbol{x})$ over $\mathcal{C}$ in order to compare it with the threshold 0. Specifically, we consider an $L$ layer feed-forward neural network, $f : \mathbb{R}^{|\boldsymbol{x}|} \to \mathbb{R}$, with ReLU activation units such that for any $\boldsymbol{x}_0 \in \mathcal{C} \subset \mathbb{R}^{|\boldsymbol{x}|}$, $f(\boldsymbol{x}_0) = \hat{\boldsymbol{x}}_L \in \mathbb{R}$ where

$$\hat{\boldsymbol{x}}_{i+1} = W^{i+1}\boldsymbol{x}_i + \boldsymbol{b}^{i+1}, \qquad \text{for } i = 0, \dots, L-1, \tag{2a}$$

$$\boldsymbol{x}_i = \max(\hat{\boldsymbol{x}}_i, 0), \qquad \text{for } i = 1, \dots, L-1. \tag{2b}$$

The terms $W^i$ and $\boldsymbol{b}^i$ refer to the weights and biases of the $i$-th layer of the neural network f. Domain $\mathcal{C}$ can be an $\ell_p$ norm ball with radius $\epsilon$. Finding the minimum of $f$ is a challenging task, as the optimization problem is generally NP hard (Katz et al., 2017). To deal with the inherent difficulty of the optimization problem itself, BaB (Bunel et al., 2018) is generally adopted. In detail, BaB based methods divide $\mathcal{C}$ into sub-domains, each of which defines a new sub-problem (branching). They then compute a relaxed lower bound of the minimum on each sub-problem (bounding). The minimum of the lower bounds of all the generated sub-domains constitutes a valid global lower bound of the global minimum over $\mathcal{C}$. As a recursive process, BaB keeps partitioning the sub-domains to tighten the global lower bound. The process terminates when the computed global lower bound is above zero (property is true) or when an input with a negative output is found (property is false). A detailed description of the BaB is provided in the appendices. In what follows, we provide a brief description of the two components, bounding methods and branching strategies, that is necessary for the understanding of our novel learning framework.

### 3.2 BOUNDING

For NN verification problems, bounding consists of finding upper and lower bounds for the final output, the minimum of $f(\boldsymbol{x})$ over $\mathcal{C}$. An effective technique to compute a lower bound is to transform the original optimization problem into a linear program (LP) by introducing convex relaxations over ReLU activation units. As we can see in Eq. (2b), ReLU activation units do not define a convex feasible set, and hence, relaxations are needed. Denote the $j$-th element of the vector $\boldsymbol{x}_i$ as $x_{i[j]}$.

Possible convex relaxations for a hidden node $x_{i[j]}$ that have been introduced so far are shown in Figure 1. We replace ReLU with the shaded green area. The tighter the convex relaxation introduced, the more computational expensive it is to compute a bound but the tighter the bound is going to be. From Figure 1, we note that in order to introduce a convex relaxation, we need intermediate bounds $l_{i[j]}$ and $u_{i[j]}$. Thus intermediate bounds are required for building the LP for the final output lower bound. Given their purpose and the large number of intermediate bound computations, rough estimations are mainly used. On the other hand, the final output lower bound is directly used in making the pruning decision and hence a tighter lower bound is preferred as it avoids further unnecessary splits on the sub-problem.

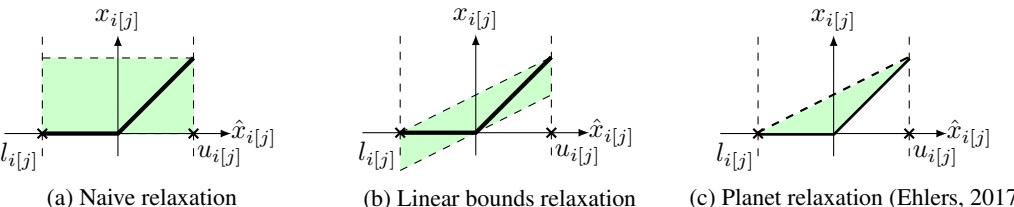

(a) Naive relaxation      (b) Linear bounds relaxation      (c) Planet relaxation (Ehlers, 2017)

Figure 1: Different convex relaxations introduced. For each plot, the black line shows the output of a ReLU activation unit for any input value between $l_{i[j]}$ and $u_{i[j]}$ and the green shaded area shows the convex relaxation introduced. Naive relaxation (a) is the loosest relaxation. Linear bounds relaxation (b) is tighter and is introduced in Weng et al. (2018). Finally, Planet relaxation (c) is the tightest linear relaxation among the three considered (Ehlers, 2017). Among them, (a) and (b) have closed form solutions which allow fast computations while (c) requires an iterative procedure to obtain an optimal solution.

## 3.3 BRANCHING

Branching is of equal importance as bounding in the BaB framework. Especially for large scale networks $f$, each branching step has a large number of putative choices. In these cases, the effectiveness of a branching strategy directly determines the possibility of verifying properties over these networks within a given time limit. On neural networks, two types of branching decisions are used: input domain split and hidden activation unit split.

Assume we want to split a parent domain $\mathcal{D}$. Input domain split selects an input dimension and then makes a cut on the selected dimension while the rest of the dimensions remain the same. The common choice is to cut the selected dimension in half and the dimension to cut is decided by the branching strategy used. Available input domain split strategies are Bunel et al. (2018) and Royo et al. (2019). Royo et al. (2019)'s is based on sensitivity test of the LP on $\mathcal{D}$ while Bunel et al. (2018) use the formula provided in Wong & Kolter (2018) to estimate final output bounds for sub-domains after splitting on each input dimension and selects the dimension that results in the highest output lower bound estimates.

In our setting, we refer to a ReLU activation unit $x_{i[j]} = \max(\hat{x}_{i[j]}, 0)$ as ambiguous over $\mathcal{D}$ if the upper bound $u_{i[j]}$ and the lower bound $l_{i[j]}$ for $\hat{x}_{i[j]}$ have different signs. Activation unit split chooses among ambiguous activation units and then divides the original problem into cases of different activation phase of the chosen activation unit. If a branching decision is made on $x_{i[j]}$, we divide the ambiguous case into two determinable cases: $\{x_{i[j]} = 0, l_{i[j]} \leq \hat{x}_{i[j]} \leq 0\}$ and $\{x_{i[j]} = \hat{x}_{i[j]}, 0 \leq \hat{x}_{i[j]} \leq u_{i[j]}\}$. After the split, the originally introduced convex relaxation is removed, since the above sets are themselves convex. We expect large improvements on the output lower bounds of the newly generated sub-problems if a good branching decision is made. Apart from random selection, employed in Ehlers (2017) and Katz et al. (2017), available ReLU split heuristics are Wang et al. (2018a) and Bunel et al. (2019). Wang et al. (2018a) compute scores based on gradient information to prioritise ambiguous ReLU nodes. Similarly, Bunel et al. (2019) use scores to rank ReLU nodes but scores are computed with a formula developed on the estimation equations in Wong & Kolter (2018). We note that for both branching strategies, after the split, intermediate bounds are updated accordingly on each new sub-problem. For NN verification problems, either domain split or ReLU split can be used at each branching step. When compared with each other, ReLU split is a more effective choice for large scale networks, as shown in Bunel et al. (2019).

All the aforementioned existing branching strategies use hand-designed heuristics. In contrast, we propose a new framework for branching strategies by utilizing a GNN to learn to imitate strong branching heuristics. This allows us to harness the effectiveness of strong branching strategies while retaining the efficiency of GPU computing power.

## 4 GNN FRAMEWORK

**Overview**  We begin with a brief overview of our overall framework, followed by a detailed description of each of its components. A graph neural network $G$ is represented by two components: a set of nodes $V$ and a set of edges $E$, such that $G = (V, E)$. Each node and each edge has its set of features. A GNN uses the graph structure and node and edge features to learn a representation vector (embedding vector) for each node $v \in V$. The GNN is a key component of our framework, in which we treat the neural network $f$ as a graph $G_f$. A GNN takes $G_f$ as an input and initializes an embedding vector for each node in $V$. The GNN updates each node's embedding vector by aggregating its own node features and all its neighbours' embedding vectors. After several rounds of updates, we obtain a learned representation (an updated embedding vector) of each node. To make a branching decision, we treat the updated embedding vectors as inputs to a score function, which assigns a score for each node that constitutes a potential branching option. A branching decision is made based on the scores of potential branching decision nodes. Our framework is visualised in Figure 2. We now describe each component in detail.

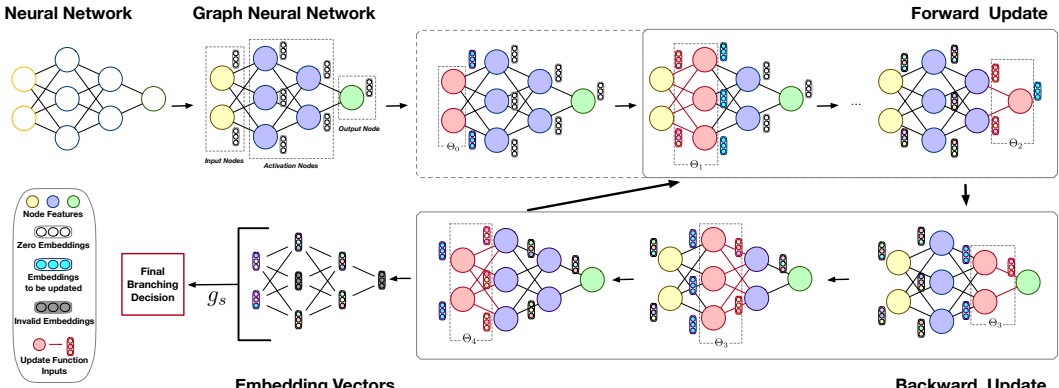

Figure 2: Illustration of our proposed GNN framework. An all zeros embedding network mimicking the neural network is initialised. Embedding vectors are updated via several rounds of forward backward passes using updating Eqs. (3)-(7). We obtain the final branching decision by calling a score function $g_s$ over all embedding vectors of the potential branching decision nodes.

**Nodes**  Given a neural network $f$, $V$ consists of all input nodes $v_{0[j]}$, all hidden activation nodes $v_{i[j]}$ and an output node $v_L$. In our framework, we combine every pre-activation variable and its associated post-activation variable and treat them as a single node. Pre- and post-activation nodes together contain the information about the amount of convex relaxation introduced at this particular activation unit, so dealing with the combined node simplifies the learning process. In terms of the Eq. (2), let $x'_{i[j]}$ denote the combined node of $\hat{x}_{i[j]}$ and $x_{i[j]}$. The nodes $v_{0[j]}$, $v_{i[j]}$ and $v_L$ are thus in one-to-one correspondence with $x_{0[j]}$, $x'_{i[j]}$ and $x_L$. We note that $V$ is larger than the set of all potential branching decisions as it includes unambiguous activation nodes and output nodes.

**Node features**  Different types of nodes have different sets of features. In particular, input node features contain the corresponding domain lower and upper bounds and the primal solution. For activation nodes, the node features consist of associated intermediate lower and upper bounds, the layer bias, primal and dual solutions and new terms computed using previous features. Finally, the output node has features including the associated output lower and upper bounds, the layer bias and the primal solution. Other types of features could be used and some features could be excluded if they are expensive to compute. We denote input node features as $z_{0[j]}$, activation node features as $z_{i[j]}$ and output node features as $z_L$. Our framework uses simple node features and does not rely on extensive feature engineering. Nonetheless, by relying on the powerful GNN framework, it provides highly accurate branching decisions.

**Edges**  $E$ consists of all edges connecting nodes in $V$, which are exactly the connecting edges in $f$. Edges are characterized by the weight matrices that define the parameters of the network $f$ such that for an edge $e^i_{jk}$ connecting $x'_{i[j]}$ and $x'_{i+1[k]}$, we assign $e^i_{jk} = W^i_{jk}$.

**Embeddings**  We associate a $p$-dimensional embedding vector $\boldsymbol{\mu}_v$ for each node $v \in V$. All embedding vectors are initialised as zero vectors.

**Forward and Backward embedding updates**   In general, a graph neural network learns signals from a graph by acting as a function of two inputs: a feature matrix $\mathcal{X} \in \mathbb{R}^{|V| \times p}$, where each row is the embedding vector $\boldsymbol{\mu}_v$ for a node $v \in V$, and an adjacency matrix $\mathcal{A}$ representing the graph structure. Under this formulation, all node embedding vectors are updated at the same time and there is no particular order between nodes. In this work, instead, we propose an update scheme where only the nodes corresponding to the same layer of the network $f$ are updated at the same time, so embedding vector updates are carried out in a layer-by-layer forward-backward way.

We argue that the forward-backward updating scheme is a natural fit for our problem. In more detail, for a given problem $\mathcal{D}$, each branching decision (an input node or an ambiguous activation node) will generate two sub-problems $s_1$ and $s_2$, with each sub-domain having an output lower bound $l_{s_1}^L$ and $l_{s_2}^L$ respectively, equal to or higher than $l_{\mathcal{D}}^L$ the lower bound that of $\mathcal{D}$. Strong branching heuristic uses a predetermined function to measure the combined improvement of $l_{s_1}^L$ and $l_{s_2}^L$ over $l_{\mathcal{D}}^L$ and makes the final branching decision by selecting the node that gives the largest improvement. Thus, to maximise the performance of a graph neural network, we want a node embedding vector to maximally capture all information related to the computation of $l_{s_1}^L$ and $l_{s_2}^L$. For estimating $l_{s_1}^L, l_{s_2}^L$ of splitting on a potential branching decision node $v$, we note that these values are closely related to two factors. The first factor is the amount of convex relaxations introduced at a branching decision node $v$, when $v$ corresponds to an ambiguous activation node. The second factor considers that the impact that splitting node $v$ will have on the convex relaxations introduced to nodes on layers after that of $v$. Recall that, if there are no ambiguous activation nodes, the neural network $f$ is simply a linear operator, whose minimum value can be easily obtained. When ambiguous activation nodes are present, the total amount of relaxation introduced determines the tightness of the lower bound to $f$. We thus treat embedding vectors as a measure of local convex relaxation and its contribution to other nodes' convex relaxation.

As shown in Figure 1, at each ambiguous activation node $x'_{i[j]}$, the area of convex relaxation introduced is determined by the lower and upper bounds of the pre-activate node $\hat{x}_{i[j]}$. We observe that intermediate lower and upper bounds of a node $\hat{x}_{i[j]}$ are significantly affected by the layers prior to it and have to be computed in a layer-by-layer fashion. Based on the observation, we utilise a forward layer-by-layer update on node embedding vectors. This should allow these embedding vectors to capture the local relaxation information. In terms of the impact of local relaxation change to that of other nodes, we note that by splitting an ambiguous node into two fixed cases, all intermediate bounds of nodes on later layers will be affected, leading to relaxation changes at those nodes. We thus employ a backward layer-by-layer update to account for the impact the local change has over other nodes. Theoretically, by fixing an ambiguous ReLU node, intermediate bounds of nodes at previous layers and on the same layer might change as well. For a naturally trained neural network, the changes for these nodes should be relatively small compared to nodes on the later layers. To account for these changes, we rely on multiple rounds of forward-and-backward updates.

In summary, during the forward update, for $i = 1, \ldots, L-1$, we have, for all possible $j$,

$$\boldsymbol{\mu}_{0[j]} \longleftarrow F_{inp}(\boldsymbol{z}_{0[j]}; \boldsymbol{\theta}_0), \quad \text{if } \boldsymbol{\mu}_{0[j]} = \mathbf{0}, \tag{3}$$

$$\boldsymbol{\mu}_{i[j]} \longleftarrow F_{act}(\boldsymbol{z}_{i[j]}, \boldsymbol{\mu}_{i-1}, e^i; \boldsymbol{\theta}_1), \qquad (4) \qquad \boldsymbol{\mu}_L \longleftarrow F_{out}(\boldsymbol{z}_L, \boldsymbol{\mu}_{L-1}, e^L; \boldsymbol{\theta}_2). \tag{5}$$

During the backward update, for $i = L-1, \ldots, 1$, we have

$$\boldsymbol{\mu}_{i[j]} \longleftarrow B_{act}(\boldsymbol{z}_{i[j]}, \boldsymbol{\mu}_{i+1}, e^{i+1}; \boldsymbol{\theta}_3), \quad (6) \qquad \boldsymbol{\mu}_{0[j]} \longleftarrow B_{inp}(\boldsymbol{z}_{0[j]}, \boldsymbol{\mu}_1, e^1; \boldsymbol{\theta}_4). \tag{7}$$

Update functions $F$ and $B$ take the form of multi-layered fully-connected networks with ReLU activation functions or composites of these simple update networks. The terms $\boldsymbol{\theta}_i$ denote the parameters of the networks. A detailed description of update functions is provided in the appendices.

We point out that our forward-backward update scheme does not depend on the underlying neural network structure and thus should be generalizable to network architectures that differ from the one we use for training. However, it does rely on the information used to compute convex relaxations, so underlying data distribution, features and bounding methods are assumed to be the same when the trained model is applied to different networks. Furthermore, our forward-backward update is memory efficient, as we are dealing with one layer at a time and only the updated embedding vectors of the layer are used to update the embedding vectors in the next (forward-pass) and the previous (backward-pass) layer. This makes it readily applicable to large networks.

**Scores** At the end of the forward-backward updates, embedding vectors for potential branching decision nodes (all input nodes and ambiguous activation nodes) are gathered and treated as inputs of a score function $g_s(\cdot; \boldsymbol{\theta}_5) : \mathbb{R}^p \to \mathbb{R}$, which takes the form of a fully-connected network with parameters $\boldsymbol{\theta}_5$. It assigns a scalar score for each input embedding vector. The final branching decision is determined by picking the node with the largest score.

## 5 PARAMETER ESTIMATION

**Training** We train a GNN via supervised learning. To estimate $\boldsymbol{\Theta} := (\boldsymbol{\theta}_0, \boldsymbol{\theta}_1, \boldsymbol{\theta}_2, \boldsymbol{\theta}_3, \boldsymbol{\theta}_4, \boldsymbol{\theta}_5)$, we propose a new hinge rank loss function that is specifically designed for our framework. Before we give details of the loss, we introduce a relative improvement measure $m$ first. Given a domain $\mathcal{D}$, for each branching decision node $v$, the two generated sub-problems have output lower bounds $l_{s_1}^L$ and $l_{s_2}^L$. We measure the relative improvement of splitting at the node $v$ over the output lower bound $l_{\mathcal{D}}^L$ as follows

$$m_v := (\min(l_{s_1}^L, 0) + \min(l_{s_2}^L, 0) - 2 \cdot l_{\mathcal{D}}^L)/(-2 \cdot l_{\mathcal{D}}^L). \tag{8}$$

Intuitively, $m$ ($0 \le m \le 1$) measures the average relative sub-problem lower bound improvement to the maximum improvement possible, that is $-l_{\mathcal{D}}^L$. Any potential branching decision node $v$ can be compared and ranked via its relative improvement value $m_v$. Since we are only interested in branching nodes with large improvement measures, ranking loss is a natural choice. A direct pairwise rank loss might be difficult to learn for NN verification problems, given the large number of branching decision nodes on each domain $\mathcal{D}$. In addition, many branching decisions may give similar performance, so it is redundant and potentially harmful to the learning process if we learn a ranking among these similar nodes. To deal with these issues, we develop our loss by first dividing all potential branching nodes into $M$ classes ($M$ is much smaller than the total number of branching decision nodes) through the improvement value $m_v$ of a node. We denote the class label as $Y_v$ for a node $v$. Labels are assigned in an ascending order such that $Y_v >= Y_{v'}$ if $m_v > m_{v'}$. We then compute the pairwise hinge-rank loss on these newly assigned labels as

$$loss_{\mathcal{D}}(\boldsymbol{\Theta}) = \frac{1}{K} \sum_{i=1}^{N} \left( \sum_{j=1}^{N} \phi(g_s(\boldsymbol{\mu}_j; \boldsymbol{\Theta}) - g_s(\boldsymbol{\mu}_i; \boldsymbol{\Theta})) \cdot \mathbf{1}_{Y_j > Y_i} \right), \tag{9}$$

where $\phi(z) = (1 - z)_+$ is the hinge function, $N$ is the total number of branching decision nodes and $K$ is the total number of pairs where $Y_j > Y_i$ for any branching decision nodes $v_i, v_j$. The loss measures the average hinge loss on score difference ($g_s(\boldsymbol{\mu}_j; \boldsymbol{\Theta}) - g_s(\boldsymbol{\mu}_i; \boldsymbol{\Theta})$) for all pairs of branching decision nodes $v_i, v_j$ such that $Y_j > Y_i$. Finally, we evaluate $\boldsymbol{\Theta}$ by solving the following optimization problem:

$$\boldsymbol{\Theta} = \arg\min_{\boldsymbol{\Theta}} \frac{\lambda}{2} \|\boldsymbol{\Theta}\|^2 + \frac{1}{n} \sum_{i}^{n} loss_{\mathcal{D}_i}(\boldsymbol{\Theta}), \tag{10}$$

where the $loss_{\mathcal{D}_i}$ is the one introduced in Eq. (9) and $n$ is the number of training samples.

**Fail-safe Strategy** We introduce a fail-safe strategy employed by our framework to ensure that consistent high-quality branching decisions are made throughout a BaB process. The proposed framework uses a GNN to imitate the behavior of the strong branching heuristic. Although computationally cheap, in some cases, the output decision by the learned graph neural network might be suboptimal. When this happens, it could lead to considerably deteriorated performance for two reasons. Firstly, we observed that for certain problems, which requires multiple splits to reach a conclusion on this problem, if a few low-quality branching decisions are made at the beginning or the middle stage of the branching process, the total number of splits required might increase substantially. The total BaB path is thus, to some extent, sensitive to the quality of each branching decision apart from those made near the end of the BaB process. Secondly, once a low-quality decision is made on a given problem, a decision of similar quality is likely to be made on the two newly generated sub-problems, leading to exponential decrease in performance. Features for newly generated sub-problems are normally similar to those of the parent problem, especially in the cases where the branching decision of the parent problem is made on the later layers and loose intermediate bounds are used. Thus, it is reasonable to expect the GNN fails again on the resulting sub-problems.

To deal with this issue, we keep track of the output lower bound improvement for each branching decision, as introduced in Eq. (8). We then set a pre-determined threshold parameter. If the improvement is below the threshold, a computationally cheap heuristic is called to make a branching decision. Generally, the back-up heuristic is able to give an above-threshold improvement and generate sub-problems sufficiently different from the parent problem to allow the learned GNN to recover from the next step onwards.

**Online Learning** Online learning is a strategy to fine-tune the network for a particular property after we have learnt $\Theta$. It can be seen as an extension of the fail-safe strategy employed. Every time a heuristic branching decision node $v_h$ is used instead of the node $v_{gnn}$ chosen by the GNN, we can use $v_h$ and $v_{gnn}$ to update the GNN accordingly. Since a correct GNN model should output an embedding vector $\boldsymbol{\mu}_h$ resulting in a higher score $g_s(\boldsymbol{\mu}_h; \Theta)$ for the heuristic decision, a loss can be developed based on the two scores $g_s(\boldsymbol{\mu}_h; \Theta)$ and $g_s(\boldsymbol{\mu}_{gnn}; \Theta)$ to generate optimization signals for correcting the GNN behaviour. For example, the loss used in our experimental setting is:

$$loss_{online}(\Theta) = g_s(\boldsymbol{\mu}_{gnn}; \Theta) - g_s(\boldsymbol{\mu}_h; \Theta) + \gamma \cdot ((m_h - m_{gnn}) > t). \tag{11}$$

The last term is used to amplify ($\gamma > 0$) the loss if the relative improvement made by the heuristic decision is more than $t$ percent higher than that by the GNN. We update $\Theta$ of the GNN by taking one gradient step with a small learning rate of the following minimization problem.

$$\Theta = \arg\min_{\Theta} \frac{\lambda}{2}\|\Theta\|^2 + loss_{online}(\Theta). \tag{12}$$

Online learning is property specific: it uses the decisions made by heuristics to fine tune the GNN model so it can best accommodate the property at hand. As will be shown in our experiments, a small but significant improvement in performance is achieved when online learning is used.

## 6 EXPERIMENTS

We now validate the effectiveness of our proposed framework through comparative experiments against other available NN verification methods. A comprehensive study of NN verification methods has been done in Bunel et al. (2019). We thus design our experiments based on the results presented in Bunel et al. (2019).

### 6.1 SETUP

We are interested in verifying properties on large network architectures with convolutional layers. In Bunel et al. (2019), existing NN methods are compared on a robustly trained convolutional network on MNIST. We adopt a similar network structure but using a more challenging dataset, namely CIFAR-10, for an increased difficulty level. We compare against the following two methods: (i) MIPplanet, a mixed integer solver backed by the commercial solver Gurobi; and (ii) BaBSR, a BaB based method utilising a ReLU-split heuristic. Our choice is motivated by their superior performance over other methods for MNIST verification problems in the previous work (Bunel et al., 2019).

We provide the detailed experimental setup through four perspectives: bounding methods, branching strategies, network structures, and verification properties tested. (**Bounding methods**) We compute intermediate bounds using linear bounds relaxations (Figure 1(b)). For the output lower bound, we use Planet relaxation (Figure 1(c)) and solve the corresponding LP with Gurobi. For the output upper bound, we compute it by directly evaluating the network value at the input provided by the LP solution. (**Branching strategy**) We focus on ReLU split only in our experiments. As shown in Bunel et al. (2019), domain split only outperforms ReLU split on low input dimensional and small scale networks. Also, since one of the Baseline method BaBSR employs a ReLU-split heuristic, we consider ReLU split only for a fair comparison. However, we emphasize that our framework is readily applicable to work with a combined domain and ReLU split strategy. (**Network structures**) Three neural network structures will be studied. The base one is of the similar structure and size to the one used in Bunel et al. (2019). It has two convolutional layers, followed by two fully connected layers and is trained robustly using the method provided in Wong & Kolter (2018). This particular choice of network size is made because the time required for solving each LP increases substantially with the size of the network. To best evaluate the performance of the branching strategy, we have to work with a medium sized network so that within the given timeout, a sufficient amount of branching decisions can be made to allow effective comparisons. When testing the transferability of the framework, two larger networks will be tested but their sizes are still restricted by the LP bottleneck. A detailed description of the network architecture is provided in the appendices. (**Verification properties**) Finally, we consider the following verification properties. Given an image $\boldsymbol{x}$ for which the model correctly predicted the label $y_c$, we randomly choose a label $y_{c'}$ such that for a given $\epsilon$, we want to prove $(\boldsymbol{e}^{(c)} - \boldsymbol{e}^{(c')})^T f'(\boldsymbol{x}') > 0$, $\forall \boldsymbol{x}'$ s.t $\|\boldsymbol{x} - \boldsymbol{x}'\|_\infty \leq \epsilon$. Here, $f'$ is the original neural network, $\boldsymbol{e}^{(c)}$ and $\boldsymbol{e}^{(c')}$ are one-hot encoding vectors for labels $y_c$ and $y_{c'}$. We want to verify that for a given $\epsilon$, the trained network will not make a mistake by labelling the image as $y_{c'}$. Since BaBSR is claimed to be the best performing method on convolutional networks, we use it to determine the

$\epsilon$ values, which govern the difficulty level of verification properties. Small $\epsilon$ values mean that most ReLU activation units are fixed so their associated verification properties are easy to prove while large $\epsilon$ values could lead to easy detection of counter-examples. The most challenging $\epsilon$ values are those at which a large number of activation units are ambiguous. We use binary search with BaBSR method to find suitable $\epsilon$ values. We only consider $\epsilon$ values that result in true properties and timed out properties. Binary search process is simplified by our choice of robustly trained models. Since these models are trained to be robust over a $\delta$ ball, the predetermined value $\delta$ can be used as a starting value for binary search.

## 6.2 TRAINING DATASET

In order to generate training data, we firstly pick 565 random images and for each image, we randomly select an incorrect class. For each property, the $\epsilon$ value is determined by running binary search with BaBSR and 800 seconds timeout, so the final set of properties consists of mainly easily solvable properties and a limited number of timed out properties.

We collect training data along a BaB process for solving a verification property. At each given domain, given the large number of potential branching decisions, we perform the strong branching heuristic on a selected subset of all potential branching decisions. The subset consists of branching decisions that are estimated to be of high quality by the BaBSR heuristic and randomly selected ones, which ensure a minimum $5\%$ coverage on each layer.

To construct a training dataset that is representative enough of the whole problem space, we need to cover a large number of properties. In addition, within a BaB framework, it is important to include training data at different stages of a BaB process. However, running a complete BaB process with the strong branching heuristic for hundreds of properties is computationally expensive and considerably time consuming. We thus propose the following procedure for generating a training dataset to guarantee a wide coverage both in terms of the verification properties and BaB stages. For generated verification properties, we randomly select $25\%$ of non-timeout property to conduct a complete BaB process with the strong branching heuristic. For the rest of the properties, we try to generate at least $\mathcal{B} = 20$ training data for each verification property. Given the maximum number of branches $q = 10$ and an effective and computationally cheap heuristic, we first generate a random integer $k$ from $[0, q]$. Then, we run a BaB process with the selected cheap heuristic for $k$ steps. Finally, we call the strong branching heuristic to generate a training sample. We repeat the process until $B$ training samples are generated or the BaB process terminated. A detailed algorithm is provided in the appendices.

## 6.3 BASE MODEL

We test our learned model on the same model structure but on properties of three different difficulty levels. Testing verification properties are generated by binary search with BaBSR and 3600s timeout. We categorise verification properties solved within 800s as easy, which is consistent with training data generated, between 800s and 2400s as medium and more than 2400s as hard. In total, we generated 467 easy properties, 773 medium properties and 426 hard properties.

Results are given in the Table 1. Methods are compared in three perspectives: the average time over all properties, average number of branches required over the properties that are solved by all methods (we exclude timed out properties) and also the ratio of timed out properties. Since the properties are generated based on BaBSR, the timed out ratios of BaBSR on easy and medium properties are not comparable with that of other methods. All other numbers should give a fair evaluation of the effectiveness of our branching strategy. BaBSR, GNN and GNN-online only differ in the branching strategy used.

On all three sets of properties, we see that our learned branching strategy has led to a more than $50\%$ reduction in the total average number of branches required for a property. As a direct result, the average time required achieves at least a $50\%$ reduction as well. Our framework is thus an effective scheme and enjoys horizontal transferability. A further performance improvement is obtained through instance-specific online learning. Among all 1666 tested verification properties, GNN with online-learning solves $61.52\%$ of properties with fewer number of branches and $60.20\%$ of properties in less time when compared to the standard GNN.

We also provide a time cactus plot (Figure 3a) for all properties on the Base model. Time cactus plots for each category of properties can be found in the appendices. All these time cactus plots look

Table 1: Methods' Performance on the Base model. For easy, medium and difficult level verification properties, we compare methods' average solving time, average number of branches required and the percentage of timed out properties. GNN-Online outperforms other methods in all aspects.

| Method | Easy time(s) | Easy branches | Easy %Timeout | Medium time(s) | Medium branches | Medium %Timeout | Hard time(s) | Hard branches | Hard %Timeout |
|---|---|---|---|---|---|---|---|---|---|
| BaBSR | 545.457 | 578.828 | 0.0 | 1370.395 | 1405.301 | 0.0 | 3127.995 | 2493.870 | 0.413 |
| MIPplanet | 1499.375 | | 0.165 | 2240.980 | | 0.430 | 2250.623 | | 0.462 |
| GNN | 272.695 | 285.682 | 0.002 | 592.118 | 583.210 | 0.004 | 1577.156 | 995.437 | 0.216 |
| GNN-online | **208.821** | **252.807** | **0.002** | **556.163** | **501.595** | **0.001** | **1369.326** | **813.502** | **0.183** |

Table 2: Methods' Performance on Large Models. For verification properties on Wide large model and Deep large model, we compare methods' average solving time, average number of branches required and the percentage of timed out properties. GNN-Online outperforms other methods in all aspects.

| Method | Wide time | Wide branches | Wide %Timeout | Deep time | Deep branches | Deep %Timeout |
|---|---|---|---|---|---|---|
| BaBSR | 4137.467 | 843.476 | 0.0 | 4016.336 | 416.824 | 0.0 |
| MIPplanet | 5855.059 | | 0.743 | 5426.160 | | 0.608 |
| GNN | 2367.693 | 387.403 | 0.127 | 2308.612 | 208.760 | 0.048 |
| GNN-online | **2179.306** | **353.879** | **0.095** | **2220.351** | **199.032** | **0.040** |

similar. Although BaBSR performs better than the commercial solver encoded method MIPplanet overall, MIPplanet wins on a subset of properties. The learned model GNN, however, is capable of giving consistent high quality performance over all properties tested.

## 6.4 Transferability: larger models

We also robustly trained two larger networks. One has the same layer structure as the Base model but has more hidden units on each layer, which we refer to as the Wide model. The other has a similar number of hidden units on each layer but more layers. We refer to it as the Deep model. The detailed network architecture is provided in the appendices. Apart from the network structure, everything else is kept the same as for the Base model experiments. We use BaBSR and a timeout of 7200s to generate 300 properties for the Wide model and 250 properties for the Deep model. For these two models, each LP called for solving a sub-problem output lower bound is much more time consuming, especially for the Deep model. This is reason that the average number of branches considered is much fewer that those of the Base model within the given time limit.

The model learned on the Base network is tested on verification properties of large networks. Experimental results are given in the Table 2 and time cactus plots (Figures 3b, 3c) are also provided. All results are similar to what we observed on the Base model, which show that our framework enjoys vertical transferability.

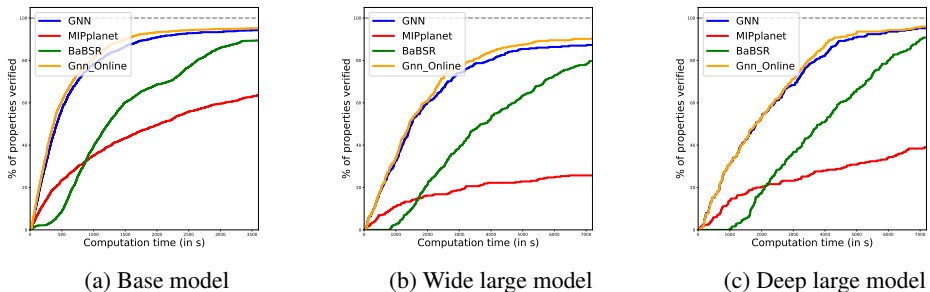

| (a) Base model | (b) Wide large model | (c) Deep large model |
|---|---|---|

Figure 3: Cactus plots for the Base model (left), Wide large model (middle) and Deep large model (right). For each model, we plot the percentage of properties solved in terms of time for each method. Consistent performances are observed on all three models. BaBSR beats MIPplanet on the majority of properties. GNN consistently outperforms BaBSR and MIPplanet. Further small improvements can be achieved through online-learning.

## 7 Discussion

The key observation of our work is that the neural network we wish to verify can be used to design a GNN to improve branching strategies. This observation can be used in enhancing the performances of other aspects of BaB. Possible future works include employing GNNs to find fast-converging starting values for solving LPs on a neural network and utilising GNNs to develop a lazy verifier, that only solves the corresponding LP on a domain when it could lead to pruning.

### Acknowledgments

This work is supported by Clarendon Fund Scholarship. We thank Rudy Bunel for useful discussions. We thank Florian Jaeckle for proofreading the article.

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

## APPENDIX A. BRANCH AND BOUND ALGORITHM

The following generic Branch and Bound Algorithm is provided in Bunel et al. (2019). Given a neural network *net* and a verification property *problem* we wish to verify, the BaB procedure examines the truthfulness of the property through an iterative procedure. During each step of BaB, we first use the *pick_out* function (line 6) to choose a problem *prob* to branch on. The split function (line 7) determines the branching strategy and splits the chosen problem *prob* into sub-problems. We compute output upper and lower bounds on each sub-problem with functions *compute_UB* and *compute_LB* respectively. Newly computed output upper bounds are used to tighten the global upper bound, which allows more sub-problems to be pruned. We prune a sub-problem if its output lower bound is greater than or equal to the global upper bound, so the smaller the global upper bound the better it is. Newly calculated output lower bounds are used to tighten the global lower bound, which is defined as the minimum of the output lower bounds of all remained sub-problems after pruning. We consider the BaB procedure converges when the difference between the global upper bound and the global lower bound is smaller than $\epsilon$.

In our case, our interested verification problem Eq. (1) is a satisfiability problem. We thus can simplify the BaB procedure by initialising the global upper bound *global_ub* as $0$. As a result, we prune all sub-problems whose output lower bounds are above $0$. In addition, the BaB procedure is terminated early when a below $0$ output upper bound of a sub-problem is obtained, which means a counterexample exits.

---

**Algorithm 1** Branch and Bound

---

1: **function** $\text{BAB}(\text{net}, \text{problem}, \epsilon)$
2:    $\text{global\_lb} \leftarrow \text{compute\_LB}(\text{net}, \text{problem})$
3:    $\text{global\_ub} \leftarrow \text{compute\_UB}(\text{net}, \text{problem})$
4:    $\text{probs} \leftarrow [(\text{global\_lb}, \text{problem})]$
5:    **while** $\text{global\_ub} - \text{global\_lb} > \epsilon$ **do**
6:       $(\_, \text{prob}) \leftarrow \text{pick\_out}(\text{probs})$
7:       $[\text{subprob\_1}, \ldots, \text{subprob\_s}] \leftarrow \text{split}(\text{prob})$
8:       **for** $i = 1 \ldots s$ **do**
9:          $\text{sub\_lb} \leftarrow \text{compute\_LB}(\text{net}, \text{subprob\_i})$
10:         $\text{sub\_ub} \leftarrow \text{compute\_UB}(\text{net}, \text{subprob\_i})$
11:         **if** $\text{sub\_ub} < \text{global\_ub}$ **then**
12:            $\text{global\_ub} \leftarrow \text{sub\_ub}$
13:            $\text{prune\_probs}(\text{probs}, \text{global\_ub})$
14:         **end if**
15:         **if** $\text{sub\_lb} < \text{global\_ub}$ **then**
16:            $\text{probs.append}((\text{sub\_lb}, \text{subprob\_i}))$
17:         **end if**
18:       **end for**
19:       $\text{global\_lb} \leftarrow \min\{\text{lb} \mid (\text{lb}, \text{prob}) \in \text{probs}\}$
20:    **end while**
21:    **return** $\text{global\_ub}$
22: **end function**

---

APPENDIX B. IMPLEMENTATION OF FORWARD AND BACKWARD PASSES

We give implementation details of forward and backward updates for embedding vectors for the model used in the experiments section. Choices of forward and backward update functions are based on the bounding methods used. In our experiments, we used linear bound relaxations for computing intermediate bounds and Planet relaxation for computing the final output lower bound. We start with a graph neural network mimicking the structure of the network we want to verify. We denote domain lower and upper bounds as $\boldsymbol{l}_0$ and $\boldsymbol{u}_0$ respectively. Similarly, we denote the intermediate bounds (pre-activation) for layers $i = 1, \ldots, L-1$ as $\boldsymbol{l}_i$ and $\boldsymbol{u}_i$. Since an LP solver is called for the final output lower bound, we have primal values for all nodes of $V$ and dual values for all ambiguous nodes of $V$. Finally, let $W^1, \ldots, W^L$ be the layer weights and $\boldsymbol{b}^1, \ldots, \boldsymbol{b}^L$ be the layer biases of the network $f$, which we wish to verify.

B.1 FORWARD PASS

Unless otherwise stated, all functions $F_*$ are 2-layer fully connected network with ReLU activation units.

B.1.1 INPUT NODES

We update the embedding vectors of input nodes only during the first round of forward pass. That is we update $\boldsymbol{\mu}_{0[j]}$ when it is zero for all $j$. After that, input nodes embedding vectors are updated only in backward pass. For each input node, we form the feature vector $\boldsymbol{z}_{0[j]}$ as a vector of $l_{0[j]}$, $u_{0[j]}$ and its associated primal solution. The input node embedding vectors are computed as

$$\boldsymbol{\mu}_{0[j]} = F_{inp}(\boldsymbol{z}_{0[j]}; \boldsymbol{\theta}_0). \tag{13}$$

B.1.2 ACTIVATION NODES

The update function $F_{act}$ can be broken down into three parts: 1) compute information from local features 2) compute information from neighbourhood embedding vectors and 3) combine information from 1) and 2) to update current layer's embedding vectors.

**Information from local features** Since we compute the final lower bound with the Planet relaxation (Figure 1(c)), we introduce a new feature related to the relaxation: the intercept of the relaxation triangle, shown in Figure 4. We denote an intercept as $\beta$ and compute it as

$$\beta_{i[j]} = \frac{-l_{i[j]} \cdot u_{i[j]}}{u_{i[j]} - l_{i[j]}}. \tag{14}$$

The intercept of a relaxation triangle can be used as a measure of the amount of relaxation introduced at the current ambiguous node.

Therefore, the local feature vector $\boldsymbol{z}_{i[j]}$ of an ambiguous node $x'_{i[j]}$ consists of $l_{i[j]}$, $u_{i[j]}$, $\beta_{i[j]}$, its associated layer bias value, primal values (one for pre-activation variable and one for post-activation variable) and dual values. We obtain information from local features via

$$R_{i[j]} = \begin{cases} F_{act-lf}(\boldsymbol{z}_{i[j]}; \boldsymbol{\theta}_1^0) & \text{if } x'_{i[j]} \text{ is ambiguous,} \\ \boldsymbol{0} & \text{otherwise.} \end{cases} \tag{15}$$

where $R_{i[j]} \in \mathbb{R}^p$.

**Information from neighbourhood embedding vectors** During the forward pass, we focus on embedding vectors of the previous layer only. To update an embedding vector on layer $i$, we first combine embedding vectors of the previous layer with edge weights via

$$E_{i[j]} = \sum_k W_{kj}^i \cdot \boldsymbol{\mu}_{i-1[k]}. \tag{16}$$

To compute the information from neighbourhood embedding vectors to an arbitrary activation node $x'_{i[j]}$, we consider each activation unit as a *gate*. We observe that the amount of the information

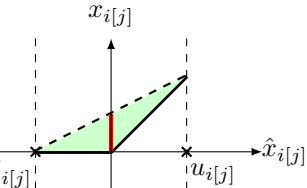

Figure 4: Red line represents the intercept of the convex relaxation. It is treated as a measure of the shaded green area.

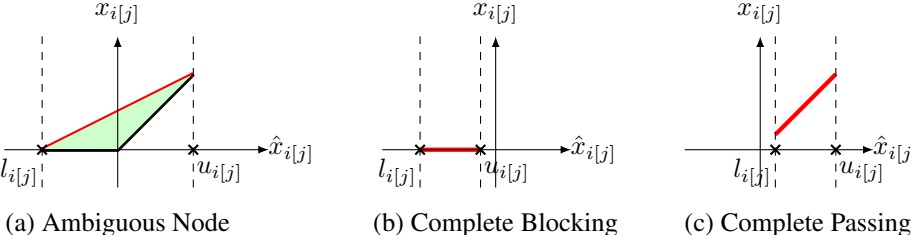

(a) Ambiguous Node      (b) Complete Blocking      (c) Complete Passing

Figure 5: Depending on the value of $l_{i[j]}$ and $u_{i[j]}$, relaxed activation function can take three forms. The left figure shows the case where $l_{i[j]}$ and $u_{i[j]}$ are of different signs. In this case, for any input value between $l_{i[j]}$ and $u_{i[j]}$, the maximum output achievable is indicated by the red line. The middle figure shows the case where both $l_{i[j]}$ and $u_{i[j]}$ are no greater than zero. In this case, the activation function completely blocks all input information by outputting zero for any input value. The right figure shows the case where $l_{i[j]}$ and $u_{i[j]}$ are greater or equal to zero. In this case, the activation function allows complete information passing by outputting a value equal to the input value.

from neighbourhood embedding vectors that remains after passing through a *gate* is dependent on the its lower bound $l_{i[j]}$ and upper bound $u_{i[j]}$. When $l_{i[j]}$ and $u_{i[j]}$ are of different signs, $x'_{i[j]}$ is an ambiguous node. With relaxation, for any input value between $l_{i[j]}$ and $u_{i[j]}$, the maximum output achievable after passing an activation unit is shown by the red slope in Figure 5(a). The red slope $s_{i[j]}$ is computed as

$$s_{i[j]}(\hat{x}_{i[j]}) = \frac{u_{i[j]}}{u_{i[j]} - l_{i[j]}} \cdot \hat{x}_{i[j]} + \beta_{i[j]}. \tag{17}$$

Thus, the amount of information from neighbourhood embedding vectors that remains after passing through an ambiguous *gate* is related to the ratio $\alpha := \frac{u_{i[j]}}{u_{i[j]} - l_{i[j]}}$. When $u_{i[j]}$ is no greater than zero, the activation node $x'_{i[j]}$ completely blocks all information. For any input value, the output value is zero after passing the activation unit, as shown by the red line in Figure 5(b). We have $\alpha = 0$ in this case. Finally, when $l_{i[j]}$ is no less than 0, the activation node $x'_{i[j]}$ allows a complete passing of information and $\alpha = 1$. It is shown by the red line in Figure 5(c). We incorporate these observations into our evaluations and compute the information from neighbourhood embedding vectors as

$$N_{i[j]} = f_{act-nb}([\alpha \cdot E_{i[j]}, \alpha' \cdot E_{i[j]}]; \boldsymbol{\theta}_1^1), \tag{18}$$

where $\alpha' = 1 - \alpha$ when $0 < \alpha < 1$ and $\alpha' = \alpha$ otherwise. Here, we use $[\boldsymbol{a}, \boldsymbol{b}]$ to denote the concatenation of two vectors $\boldsymbol{a}, \boldsymbol{b} \in \mathbb{R}^p$ into a vector of $\mathbb{R}^{2p}$. We introduce $\alpha'$ to be more informative. We do not consider the information that relate to the intercept $\beta_{i[j]}$ in the ambiguous case for the sake of simplicity. Improved performance could be expected if the $\beta_{i[j]}$ related information is incorporated as well.

**Combine previous information** Finally, we combine the information from local features and the information from neighbourhood embedding vectors to update the embedding vectors of activation nodes. Specifically,

$$\boldsymbol{\mu}_{i[j]} = F_{act-com}([R_{i[j]}, N_{i[j]}]; \boldsymbol{\theta}_1^2). \tag{19}$$

### B.1.3 OUTPUT NODE

Embedding vectors of output nodes are updated in a similar fashion to that of activation nodes. We first compute information from local features.

$$R_{Lj} = F_{out-lf}(\boldsymbol{z}_{Lj}; \boldsymbol{\theta}_2^0) \tag{20}$$

For output nodes, the vector of local features $\boldsymbol{z}_L$ consists of output lower bound, output upper bound, primal solution and layer bias. $F_{out-lf}$ is a one-layer fully-connected network with ReLU activation units. We then compute information from neighbourhood embedding vectors. Since the output node does not have an activation unit associated with it, we directly compute the information of neighbourhood embedding vectors as

$$E_{L[j]} = \sum_k W_{kj}^L \cdot \boldsymbol{\mu}_{L-1[k]}. \tag{21}$$

Finally, we update the embedding vector of the output node as

$$\boldsymbol{\mu}_{Lj} = F_{out-com}([R_{L[j]}, E_{L[j]}]; \boldsymbol{\theta}_2^1). \tag{22}$$

### B.2 BACKWARD PASS

During backward message passing, for $i = L-1, \ldots, 1$, we update embedding vectors for activation nodes and input node. Again, all functions $B_*$ are 2-layer fully-connected networks unless specified otherwise.

### B.2.1 ACTIVATION NODES

Similar to updates of embedding vectors carried out for activation nodes in a forward pass, we update embedding vectors of activation nodes using the same three steps in the backward pass, but with minor modifications.

**Information from local features** We use the same feature $\boldsymbol{z}_{i[j]}$ as the one used in the forward pass and compute the information from local features as

$$R_{i[j]}^b = \begin{cases} B_{act-lf_1}(\boldsymbol{z}_{i[j]}; \boldsymbol{\theta}_3^0) & \text{if } x'_{i[j]} \text{ is ambiguous,} \\ \boldsymbol{0} & \text{otherwise.} \end{cases} \tag{23}$$

We recall that a dual value indicates how the final objective function is affected if its associated constraint is relaxed by a unit. To better measure the importance of each relaxation to the final objective function, we further update the information from local features by

$$R_{i[j]}^{b'} = \begin{cases} B_{act-lf_2}([\boldsymbol{d}_{i[j]} \odot R_{i[j]}^b, R_{i[j]}^b]; \boldsymbol{\theta}_3^1) & \text{if } R_{i[j]}^b \neq \boldsymbol{0} \\ \boldsymbol{0} & \text{otherwise.} \end{cases} \tag{24}$$

Here, $\boldsymbol{d}_{i[j]}$ is the vector of dual values corresponding to the activation node $x'_{i[j]}$. We use $\odot$ to mean that we multiply $R_{i[j]}^b$ by each element value of $\boldsymbol{d}_{i[j]}$ and concatenate them as a singe vector.

**Information from neighbourhood embedding vectors** During the backward pass, we focus on embedding vectors of the next layer only. In order to update an embedding vector on layer $i$, we compute the neighbourhood embedding vectors as

$$E_{i[j]}^b = \sum_k W_{jk}^{i+1} \cdot \boldsymbol{\mu}_{i+1[k]}. \tag{25}$$

We point out that there might be an issue with computing $E_{i[j]}$ if the layer $i+1$ is a convolutional layer in the backward pass. For a convolutional layer, depending on the padding number, stride number and dilation number, each node $x'_{i[j]}$ may connect to a different number of nodes on the layer $i+1$. Thus, to obtain a consistent measure of $E_{i[j]}$, we divide $E_{i[j]}$ by the number of connecting node on the layer $i+1$, denoted as $E_{i[j]}^{b'}$ and use the averaged $E_{i[j]}^{b'}$ instead. Let

$$E_{i[j]}^{b*} = \begin{cases} E_{i[j]}^{b'} & \text{if layer i+1 convolutional,} \\ E_{i[j]}^b & \text{otherwise.} \end{cases} \tag{26}$$

The following steps are the same as the forward pass. We first evaluate

$$N_{i[j]}^b = B_{act-nb}([\alpha \cdot E_{i[j]}^{b*}, \alpha' \cdot E_{i[j]}^{b*}]; \boldsymbol{\theta}_3^2), \tag{27}$$

and the update embedding vectors as

$$\boldsymbol{\mu}_{i[j]} = B_{act-com}([R_{i[j]}^{b'}, N_{i[j]}^b]; \boldsymbol{\theta}_3^3). \tag{28}$$

### B.2.2 INPUT NODES

Finally, we update the input nodes. We use the feature vector $\boldsymbol{z}_0^b$, which consists of domain upper bound and domain lower bound. Information from local features is evaluated as

$$R_{0j} = B_{inp-lf}(\boldsymbol{z}_{0[j]}^b; \boldsymbol{\theta}_4^0). \tag{29}$$

We compute the information from neighbourhood embedding vectors in the same manner as we do for activation nodes in the backward pass, shown in Eq (26). Denote the computed information as $E_{0[j]}^{b*}$. The embedding vectors of input nodes are updated by

$$\boldsymbol{\mu}_{0[j]} = B_{inp-com}([R_{0[j]}^{b'}, E_{0[j]}^{b*}]; \boldsymbol{\theta}_4^1). \tag{30}$$

## APPENDIX C. ALGORITHM FOR GENERATING TRAINING DATASET

Algorithm 2 outlines the procedure for generating the training dataset. The algorithm ensures the generated training date have a wide coverage both in terms of the verification properties and BaB stages while at the same time is computationally efficient. Specifically, we randomly pick 25% of all properties that do not time out and run a complete BaB procedure on each of them with the strong branching heuristic to generate training samples (line 3-5). For the remaining properties, we attempt to generate $B$ training samples for each of them. To cover different stages of a BaB process of a property, we use a computationally cheap heuristic together with the strong branching heuristic. Given a property, we first use the cheap heuristic for $k$ steps (line 10-15) to reach a new stage of the BaB procedure and then call the strong branching heuristic to generate a training sample (line 16). We repeat the process until $B$ training samples are generated or the BaB processs terminates.

---

**Algorithm 2** Generating Training Dataset

---

1: Provided: total $P$ properties; minimum $B$ training data for each property; a maximum $q$ branches between strong branching decisions
2: **for** $p = 1, \ldots, P$ **do**:
3:     $\alpha \longleftarrow$ random number from $[0, 1]$
4:     **if** $p$ is not a timed out property and $\alpha \leq 0.25$ **then**
5:         Running a complete BaB process with the Strong Branching Heuristic
6:     **else**
7:         $b = 0$
8:         **while** $b \leq B$ **do**
9:             $k \longleftarrow$ random integer from $[0, q]$
10:             **while** $k > 0$ **do**
11:                 Call a computationally cheap heuristic
12:                 **if** BaB process terminates **then return**
13:                 **end if**
14:                 $k = k - 1$
15:             **end while**
16:             Call the strong branching heuristic and generate a training sample
17:             **if** BaB process terminates **then return**
18:             **end if**
19:             $b = b + 1$
20:         **end while**
21:     **end if**
22: **end for**

---

## APPENDIX D. EXPERIMENT DETAILS

All the hyper-parameters used in the experiments are determined by testing a small set of numbers over the validation set. Due to the limited number of tests, we believe better sets of hyper-parameters could be found.

### D.1 TRAINING DETAILS

**Training dataset**  To generate a training dataset, 565 random images are selected. Binary serach with BaBSR and 800 seconds timeout are used to determine $\epsilon$ on the Base model. Among 565 verification properties determined, we use 430 properties to generate 17958 training samples and the rest of properties to generate 5923 validation samples. Training samples and validation samples are generated using Algorithm 2 with $B = 20$ and $q = 10$.

For a typical epsilon value, each sub-domain generally contains 1300 ambiguous ReLU nodes. Among them, approximately 140 ReLU nodes are chosen for strong branching heuristics, which leads to roughly 200 seconds for generating a training sample. We point out that the total amount of time required for generating a training sample equals the 2*(per LP solve time)*(number of ambiguous ReLU nodes chosen). Although both the second and the third terms increase with the size of the model used for generating training dataset, the vertical transferability of our GNN enables us to efficiently generate training dataset by working with a small substitute of the model we are interested in. In our case, we trained on the Base model and generalised to Wide and Deep model.

**Training**  We initialise a GNN by assigning each node a 64-dimensional zero embedding vector. GNN updates embedding vectors through two rounds of forward and backward updates. To train the GNN, we use hinge rank loss (Eq. (9)) with $M = 10$. Parameters $\Theta$ are computed and updated through Adam optimizer with weight decay rate $\lambda = 1e^{-4}$ and learning rate $1e^{-4}$. If the validation loss does not decrease for 10 consecutive epochs, we decrease the learning rate by a factor of 5. If the validation loss does not decrease for 20 consecutive epochs, we terminate the learning procedure. The batch size is set to 2. In our experiments, each training epoch took less than 400 seconds and the GNN converges within 60 epochs.

In terms of the training accuracy, we first evaluate each branching decision using the metric defined by Eq. (8) [1]. Since there are several branching choices that give similar performance at each subdomain, we considered all branching choices that have $m_v$ above 0.9 as correct decisions. Under this assumption, our trained GNN achieves $85.8\%$ accuracy on the training dataset and $83.1\%$ accuracy on the validation dataset.

### D.2 VERIFICATION EXPERIMENT DETAILS

We ran all verification experiments in parallel on 16 CPU cores, with one property being verified on one CPU core. We observed that although we specifically set the thread number to be one for MIPplanet (backed by the commercial solver Gurobi), the time required for solving a property depends on the total number of CPUs used. For a machine with 20 cpu cores, MIPplanet requires much less time on average for proving the same set of properties on fewer (say 4) CPU cores in parallel than on many (say 16) CPU cores in parallel (the rest of CPU cores remain idle). Since BaBSR, GNN and GNN-online all use Gurobi for the bounding problems, similar time variations, depending on the number of CPU cores used, are observed. We ran each method in the same setting and on 16 CPUs in parallel, so our reported results and time are comparable. However, we remind readers to take the time variation into consideration when replicating our experiments or using our results for comparison.

**Fail-safe strategy**  Since, to the best of our knowledge, the branching heurisitc of BaBSR is the best performing one on convolutional neural networks so far, we choose it for our fail-safe strategy. The threshold is set to be 0.2. Every time when the relative improvement $m_{gnn}$ of a GNN branching decision $v_{gnn}$ is less than 0.2, we call the heuristic to make a new branching decision $v_h$. We solve

---

[1]we have tried various other metrics, including picking the minimum of the two subdomain lower bounds and the maximum of the two lower bounds. Among these metrics, metric defined by Eq. (8) performs the best.

the corresponding LPs for the new branching decision and compute its relative improvement $m_h$. The node with higher relative improvement is chosen to be the final branching decision.

**Online learning**   We take a conservative approach in terms of online learning. We refer to a GNN decision as a failed decision if the relative improvement offered by heuristic branching is better than the one offered by the GNN. We record all GNN failed decisions and only update the GNN model online when the same failed decision is made at least twice. To update the GNN model, we use Adam optimizer with weight decay rate $\lambda = 1e^{-4}$ and learning rate $1e^{-4}$. The GNN model is updated with one gradient step only with respect to the optimization problem Eq. (12), where $\gamma = 1$ and $t = 0.1$ in the loss function $loss_{online}$, defined in Eq. (11).

## D.3 BASELINES

We decided our baselines based on the experiment results of Bunel et al. (2019). In Bunel et al. (2019), methods including MIPplanet, BaBSR, planet (Ehlers, 2017), reluBaB and reluplex (Katz et al., 2017) are compared on a small convolutional MNIST network. Among them, BaBSR and MIPplanet significantly outperform other methods. We thus evaluate our methods against these two methods only in the experiments section. In order to strengthen our baseline, we compare against two additional methods here.

**Neurify (Wang et al., 2018a)**   Similar to BaBSR, Neurify splits on ReLU activation nodes. It makes a branching decision by computing gradient scores to prioritise ReLU nodes. Since the updated version of Neurify's released code supports verification, we conducted a comparison experiment between between Neurify and BaBSR for inclusiveness.

Neurify does not support CIFAR dataset. To evaluate the performance of Neurify, we obtained the trained ROBUST MNIST model and corresponding verification properties from Bunel et al. (2019). We ranked all verification properties in terms of the BaBSR solving time and selected the first 200 properties, which are solved by BaBSR within one minute, as our test properties. For a fair comparison, we have restricted Neurify to use one CPU core only and set the timeout limit to be two minutes. Among all test properties, Neurify timed out on 183 out of 200 properties. BaBSR thus outperforms Neurify significantly. Combining with the results of Bunel et al. (2019), BaBSR is indeed a fairly strong baseline to be compared against.

**MIP based algorithm (Tjeng et al., 2019)**   We also compared our MIPplanet baseline against a new MIP based algorithm (Tjeng et al., 2019), published in ICLR 2019. To test these two methods, we randomly selected 100 verification properties from the CIFAR Base experiment with timeout 3600s. In terms of solving time, MIPplanet requires 1732.18 seconds on average while the new MIP algorithm requires 2736.60 seconds. Specifically, MIPplanet outperforms the new MIP algorithm on 78 out of 100 properties. MIPplanet is therefore a strong baseline for comparison.

As a caveat, we mention that the main difference between MIPplanet and the algorithm of (Tjeng et al., 2019) is the intermediate bound computation, which is complementary to our focus. If better intermediate bounds are shown to help verification, we can still use our approach to get better branching decisions corresponding to those bounds.

## D.4 MODEL ARCHITECTURE

We provide the architecture detail of the neural networks verified in the experiments in the following table.

| Network Name | No. of Properties | Network Architecture |
|:---:|:---:|:---:|
| BASE Model | Easy: 467 Medium: 773 Hard: 426 | Conv2d(3,8,4, stride=2, padding=1) Conv2d(8,16,4, stride=2, padding=1) linear layer of 100 hidden units linear layer of 10 hidden units (Total ReLU activation units: 3172) |
| WIDE | 300 | Conv2d(3,16,4, stride=2, padding=1) Conv2d(16,32,4, stride=2, padding=1) linear layer of 100 hidden units linear layer of 10 hidden units (Total ReLU activation units: 6244) |
| DEEP | 250 | Conv2d(3,8,4, stride=2, padding=1) Conv2d(8,8,3, stride=1, padding=1) Conv2d(8,8,3, stride=1, padding=1) Conv2d(8,8,4, stride=2, padding=1) linear layer of 100 hidden units linear layer of 10 hidden units (Total ReLU activation units: 6756) |

Table 3: For each CIFAR experiment, the network architecture used and the number of verification properties tested.

## APPENDIX E. ADDITIONAL EXPERIMENT RESULTS

### E.1 FAIL-SAFE HEURISTIC DEPENDENCE

In all our experiments, we have compared against BaBSR, which employs only the fail-safe heuristic for branching. In other words, removing the GNN and using only the fail-safe heuristic is equivalent to BaBSR. The fact that GNN significantly outperforms BaBSR demonstrates that GNN is doing most of the job. To better evaluate the GNN's reliance on a fail-safe heuristic, we study the ratio of times that a GNN branching decision is used for each verification property of a given model. Results are listed in Table 4. On all three models, GNN accounts for more than $90\%$ of branching decisions employed on average, ensuring the effectiveness of our GNN framework.

Table 4: Evaluating GNN's dependence on the fail-safe strategy. Given a CIFAR model, we collected the percentage of times GNN branching decision is used and the percentage of times the fail-safe heuristic (BaBSR in our case) is employed for each verification property. We report the average ratio of all verification properties of the same model. To account for extreme cases, we also list the minimum and maximum usage ratios of the fail-safe heuristic for each model.

| Model | GNN(avg) | BaBSR(avg) | BaBSR(min) | BaBSR(max) |
|:---|:---:|:---:|:---:|:---:|
| BASE | 0.934 | 0.066 | 0.0 | 0.653 |
| WIDE | 0.950 | 0.050 | 0.0 | 0.274 |
| DEEP | 0.964 | 0.036 | 0.0 | 0.290 |

### E.2 GNN FEATURE ANALYSIS

We evaluate the importance of different features used in GNN. We note that two types of features are used in GNN. The first type (including intermediates bounds, network weights and biases) can be collected at negligible costs. The other type is LP features (primal and dual values) that are acquired by solving a strong LP relaxation, which are expensive to compute but potentially highly informative. To evaluate their effect, we trained a new GNN with LP features removed and tested the new GNN on 260 randomly selected verification properties on the Base model. Among the selected properties, 140 are categorised as easy, 70 as medium and 50 as hard. We denote the model trained on all features as GNN and the newly trained model as GNN-R (we use R to indicate reduced features).

Table 5: Measuring the importance of features used by GNN. For easy, medium and difficult level verification properties, we compare methods' average solving time, average number of branches required and the percentage of timed out properties.

| | Easy | | | Medium | | | Hard | | |
|---|---|---|---|---|---|---|---|---|---|
| Method | time(s) | branches | %Timeout | time(s) | branches | %Timeout | time(s) | branches | %Timeout |
| BaBSR | 429.589 | 641.300 | 0.0 | 1622.669 | 1504.366 | 0.0 | 2466.712 | 1931.098 | 0.0 |
| GNN | 268.592 | 319.386 | 0.0 | 724.883 | 529.070 | 0.0 | 1025.826 | 772.667 | 0.0 |
| GNN-R | 348.482 | 441.043 | 0.0 | 898.011 | 720.958 | 0.0 | 1340.559 | 967.804 | 0.0 |

From Table 5, we observe that removing primal and dual information deteriorates the GNN performance, but GNN-R still outperforms the baseline heuristic BaBSR. We believe cheap features are the most important. Depending on the cost of LP, potential users can either remove expensive LP features or train a GNN with a smaller architecture.

### E.3 MIPPLANET BRANCHING NUMBER

MIPplanet is implemented with the commercial solver Gurobi. Since Gurobi outputs internal branch number, we recorded MIPplanet branch number for a subset of verification properties for each model. In detail, we randomly selected 120 properties of various difficulty levels for the Base model and 27 properties each for the Wide and Deep model. Results are summarised in Table 6.

One key observation we made is that Gurobi branch number is not positively related to the solving time. For instance, on timed out properties of the Wide model, MIPplanet branch number varies between 1 and 7479. We suspect Gurobi performs cutting before branching, so time spent on branching varies between properties, leading to inconsistent branch number and solving time. As the result, the MIPplanet branch number is not comparable with that of BaBSR, GNN and GNN-online. This is also the reason that we did not include MIPplant branch number in Table 1 and Table 2.

Table 6: Methods' performance on randomly selected properties. We show methods' average solving time, average number of branches required and the percentage of timed out properties. We emphasize that MIPplanet branch number is not comparable with those of other methods.

| | Base | | | Wide | | | Deep | | |
|---|---|---|---|---|---|---|---|---|---|
| Method | time(s) | branches | %Timeout | time(s) | branches | %Timeout | time(s) | branches | %Timeout |
| BaBSR | 1472.508 | 1420.839 | 0.067 | 2985.199 | 918.167 | 0.111 | 3811.712 | 482.167 | 0.111 |
| MIPPLANET | 1783.800 | 3780.408* | 0.258 | 5254.134 | 2949.625* | 0.556 | 4566.080 | 4332.375* | 0.407 |
| GNN | 714.224 | 817.017 | 0.017 | 996.811 | 268.333 | 0.074 | 1893.081 | 201.500 | 0.0 |
| GNN-ONLINE | 582.738 | 642.387 | 0.008 | 949.694 | 257.667 | 0.033 | 1776.278 | 192.167 | 0.0 |

### E.4 LP SOLVING TIME AND GNN COMPUTING TIME

We mention that LP solving time is the main bottleneck for branch-and-bound based verification methods. Although both GNN evaluation time and LP solving time increase with the size of network, LP solving time grows at a significantly faster speed. For instance, in CIFAR experiments, GNN requires on average 0.02, 0.03, 0.08 seconds to make a branching decision on Base, Wide and Deep model respectively but the corresponding one LP solving time on average are roughly 1.1, 4.9, 9.6 seconds. GNN evaluation is almost negligible for large neural networks when compared to LP solving time.

### E.5 GEOMETRIC MEAN

For all our experiments, we based our analyses on the statistics of average solving time and branching number. To ensure the reported numbers are not biased by potential outliers, we measure methods' performance with the geometric mean as well and summarize results in Table 7 and Table 8. Statistics of geometric mean are consistent with that of arithmetic mean, validating the analyses of the main paper.

Table 7: Methods' Performance on the Base model. For easy, medium and difficult level verification properties, we compare methods' geometric average solving time, geometric average number of branches required and the percentage of timed out properties. GNN-Online outperforms other methods in all aspects.

| | Easy | | | Medium | | | Hard | | |
|---|---|---|---|---|---|---|---|---|---|
| Method | time(s) | branches | %Timeout | time(s) | branches | %Timeout | time(s) | branches | %Timeout |
| BaBSR | 471.547 | 468.117 | 0.0 | 1304.032 | 1236.398 | 0.0 | 3094.794 | 2271.796 | 0.413 |
| MIPplanet | 836.049 | | 0.165 | 1401.273 | | 0.430 | 1374.794 | | 0.462 |
| GNN | 180.638 | 191.514 | 0.002 | 412.880 | 417.970 | 0.004 | 985.142 | 691.599 | 0.216 |
| GNN-online | **143.968** | **180.575** | **0.002** | **399.786** | **380.463** | **0.001** | **833.343** | **605.443** | **0.183** |

Table 8: Methods' Performance on Large Models. For verification properties on Wide large model and Deep large model, we compare methods' geometric average solving time, geometric average number of branches required and the percentage of timed out properties. GNN-Online outperforms other methods in all aspects.

| | Wide | | | Deep | | |
|---|---|---|---|---|---|---|
| Method | time | branches | %Timeout | time | branches | %Timeout |
| BaBSR | 3510.960 | 699.826 | 0.0 | 3580.469 | 359.464 | 0.0 |
| MIPplanet | 4430.098 | | 0.743 | 4014.469 | | 0.608 |
| GNN | 1403.841 | 296.821 | 0.127 | 1529.542 | 158.498 | 0.048 |
| GNN-online | **1316.455** | **279.298** | **0.095** | **1502.349** | **155.523** | **0.040** |

### E.6 ADDITIONAL PLOTS

We provide cactus plots for the Base model on easy, medium and hard difficulty level properties respectively.

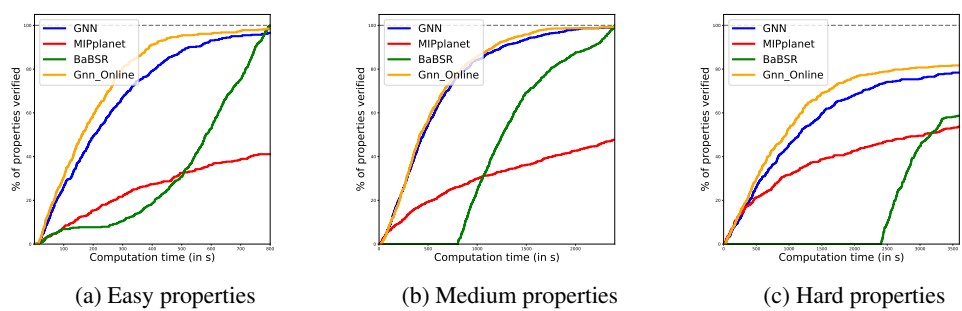

(a) Easy properties        (b) Medium properties        (c) Hard properties

Figure 6: Cactus plots for easy properties (left), medium properties (middle) and hard properties (right) on the Base model. For each category, we plot the percentage of properties solved in terms of time for each method. BaBSR beats MIPplanet on easy and medium properties overall. On hard properties, although BaBSR manages to solve more properties, its average performance is worse than MIPplanet in terms of time. GNN consistently outperforms BaBSR and MIPplanet on all three levels of properties, demonstrating the horizontal transferability of our framework. Again, further small improvements can be achieved through online-learning.

## APPENDIX F. MNIST DATASET

We replicate the CIFAR experiments on the MNIST dataset to test the generalization ability of our GNN framework.

### F.1 MODEL ARCHITECTURE AND VERIFICATION PROPERTIES

We trained three different networks on MNIST with the method provided in Wong & Kolter (2018). The base model is mainly used for generating the training dataset and testing the horizontal generalization ability of the trained GNN. The Wide and Deep models are used for evaluating the vertical generalization. Verification properties are found via binary search with BaBSR. We set the binary search time limit to be 1800 seconds for the Base model and 3600 seconds for the other two models.

| Network Name | No. of Properties | Network Architecture |
|:---:|:---:|:---:|
| BASE | 100 | Conv2d(1,4,4, stride=2, padding=1)
Conv2d(4,8,4, stride=2, padding=1)
linear layer of 50 hidden units
linear layer of 10 hidden units
(Total ReLU activation units: 1226) |
| WIDE | 100 | Conv2d(1,16,4, stride=2, padding=1)
Conv2d(16,32,4, stride=2, padding=1)
linear layer of 100 hidden units
linear layer of 10 hidden units
(Total ReLU activation units: 4804) |
| DEEP | 100 | Conv2d(1,8,4, stride=2, padding=1)
Conv2d(8,8,3, stride=1, padding=1)
Conv2d(8,8,3, stride=1, padding=1)
Conv2d(8,8,4, stride=2, padding=1)
linear layer of 100 hidden units
linear layer of 10 hidden units
(Total ReLU activation units: 5196) |

Table 9: For each MNIST experiment, the network architecture used and the number of verification properties tested.

## F.2 TRAINING DETAILS

**Training dataset** Training dataset is generated on the Base model. We point out that we explicitly choose a Base model of small network size for efficient and fast training data generation.

To generate a training dataset, 538 random MNIST images are selected. Binary serach with BaBSR and 600 seconds timeout are used to determine $\epsilon$ on the Base model. Among 538 verification properties determined, we use 403 properties to generate 18231 training samples and the rest of properties to generate 5921 validation samples. Training samples and validation samples are generated using Algorithm 2 with $B = 20$ and $q = 10$. For a typical epsilon value, each sub-domain generally contains 480 ambiguous ReLU nodes. Among them, approximately 80 ReLU nodes are chosen for strong branching heuristics, which leads to roughly 45 seconds for generating a training sample.

**Training** The same set of parameters and training procedure are used for training a GNN for MNIST dataset. The GNN converges in 70 epochs with each epoch took less than 400 seconds. The trained GNN reached $86.5\%$ accuracy on the training dataset and $83.1\%$ accuracy on the validation dataset.

## F.3 EXPERIMENT RESULTS

We first note that we use verification properties with timeout 1800 seconds on the Base model to allow for an integrated evaluation of GNN on both its performance and its horizontal transferability. Vertical transferability is tested on the Wide and Deep model.

We observe that MIPplanet outperforms all BaB based methods on verification properties of the Base model. Given that the network size of the Base model is particularly small (1226 hidden units only), we believe that MIP algorithms backed by commercial solvers could be the most effective tool on verification problems of small size. Our conjecture is further confirmed by the fact that MIPplanet timed out on almost all properties of both the Wide and Deep models. On all three models, GNN consistently outperforms BaBSR, demonstrating the transferability of our framework. Finally, when online learning is considered, we found it is effective in fine-tuning the trained GNN and enabling further performance improvements, especially on the Wide model.

Table 10: Methods' Performance on different models. For the Base, Wide and Deep model, we compare methods' average solving time, average number of branches required and the percentage of timed out properties respectively. MIPplanet performs the best on the Base model while GNN-Online outperforms other methods on the Wide and the Deep model.

| | Base | | | Wide | | | Deep | | |
|---|---|---|---|---|---|---|---|---|---|
| Method | time(s) | branches | %Timeout | time(s) | branches | %Timeout | time(s) | branches | %Timeout |
| BABSR | 878.226 | 2659.276 | 0.0 | 1888.084 | 399.571 | 0.0 | 1895.032 | 211.644 | 0.0 |
| MIPPLANET | **411.167** | | **0.0** | 3567.109 | | 0.990 | 3437.918 | | 0.833 |
| GNN | 623.291 | 1549.867 | 0.019 | 1526.381 | 322.510 | 0.040 | 1189.271 | 144.238 | 0.010 |
| GNN-ONLINE | 547.102 | **1363.057** | 0.009 | **1133.429** | **283.673** | **0.010** | **1121.066** | **139.980** | **0.0** |

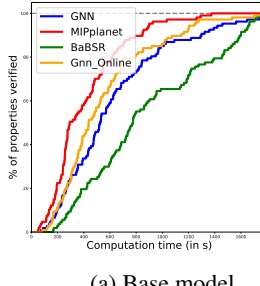
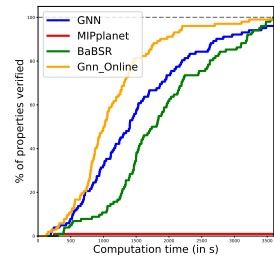
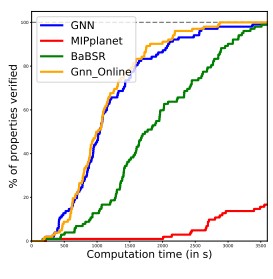

(a) Base model        (b) Wide large model        (c) Deep large model

Figure 7: Cactus plots for the Base model (left), Wide large model (middle) and Deep large model (right). For each model, we plot the percentage of properties solved in terms of time for each method. MIPplanet outperforms all BaB based method on the Base model. However, once model size get larger, MIPplanet's performance deteriorates quickly. ON the Wide and Deep model, MIPplanet timed out on most properties and is outperformed by BaBSR. GNN consistently outperforms BaBSR on all three models, demonstrating the transferability of our framework. Again, further small improvements can be achieved through online-learning.

## F.4 FAIL-SAFE HEURISTIC DEPENDENCE

Results of Table 11 ensure that the trained GNN is indeed account for the most branching decisions.

Table 11: Evaluating GNN's dependence on the fail-safe strategy. Given a MNIST model, we collected the percentage of times GNN branching decision is used and the percentage of times the fail-safe heuristic (BaBSR in our case) is employed for each verification property. We report the average ratio of all verification properties of the same model. To account for extreme cases, we also list the minimum and maximum usage ratios of the fail-safe heuristic for each model.

| Model | GNN(avg) | BaBSR(avg) | BaBSR(min) | BaBSR(max) |
|---|---|---|---|---|
| BASE | 0.962 | 0.038 | 0.0 | 0.204 |
| WIDE | 0.884 | 0.116 | 0.0 | 0.376 |
| DEEP | 0.938 | 0.062 | 0.0 | 0.407 |

## F.5 GEOMETRIC MEAN

The consistency between the results of Table 12 and Table 10 confirm that our analyses based on arithmetic mean are not biased by outliers.

Table 12: Methods' Performance on different models. For the Base, Wide and Deep model, we compare methods' geometric average solving time, geometric average number of branches required and the percentage of timed out properties respectively. MIPplanet performs the best on the Base model while GNN-Online outperforms other methods on the Wide and the Deep model.

| | Base | | | Wide | | | Deep | | |
|---|---|---|---|---|---|---|---|---|---|
| Method | time(s) | branches | %Timeout | time(s) | branches | %Timeout | time(s) | branches | %Timeout |
| BABSR | 740.740 | 2210.248 | 0.0 | 1683.816 | 362.171 | 0.0 | 1694.555 | 187.723 | 0.0 |
| MIPPLANET | 330.455 | | **0.0** | 3506.400 | | 0.990 | 3356.780 | | 0.833 |
| GNN | 501.966 | 1268.780 | 0.019 | 1267.736 | 284.452 | 0.040 | 1022.466 | 131.171 | 0.010 |
| GNN-ONLINE | 450.033 | **1123.867** | 0.009 | **973.015** | **254.641** | **0.0** | **991.552** | **128.115** | **0.0** |

## F.6 Transferability between datasets

To evaluate whether our framework can generalize further to support transferring between datasets, we tested CIFAR trained GNN on MNIST verification properties. In detail, we have tested the GNN on 20 randomly picked verification properties of MNIST Base model. We found that BABSR outperforms CIFAR trained GNN on all properties, so the CIFAR trained GNN model does not transfer to MNIST dataset. This is expected as MNIST and CIFAR images differ significantly from each other.

