# OpenReview forum: "Neural Network Branching for Neural Network Verification "
_ICLR.cc/2020/Conference — Accept (Talk)_

### Official Review · AnonReviewer3 · 2019-10-23
**Official Blind Review #3**

**Rating:** 8

**Review:**

The paper proposes learning a branching heuristic to be used inside a branch-and-bound algorithm used for solving integer programming problems corresponding to neural network verification. The heuristic is parameterized as a neural network and trained to imitate an existing heuristic called Strong Branching which is computationally expensive but produces smaller branch-and-bound trees than other heuristics. A graph neural network architecture is used to take the neural network being verified as input, and a message passing schedule that follows a forward pass and a backward pass along the network being verified is used. An online learning variant is also considered that fine tunes the learned heuristic at test time as a problem instance is being solved. Results for verifying large convolutional neural networks on CIFAR-10 show approximately 2x improvement in average running time of the branch-and-bound algorithm.

Pros:
- Significant reductions in average running time across the various datasets.
- Well-written paper with clear figures (especially figure 2) and explanations. I enjoyed reading it.

Cons:
- Novelty is somewhat low, as it is a straightforward application of existing ideas like Gasse et al. NeurIPS’19 to the problem of verification. The idea of forward and backward message passing schedule is similar to the idea considered in Amizadeh et al., ICLR’19 (https://openreview.net/pdf?id=BJxgz2R9t7).
- It would be useful to present results on other datasets like MNIST. Even if they are not as impressive, it would be good to know when the proposed approach works and when it doesn’t.

Additional comments:
- Reporting average running time and number of branches can be sensitive to outliers. Shifted geometric mean will be less sensitive, please include these metrics.
- It would be good to compare against using a mixed integer program input representation (as done in Gasse et al., NeurIPS’19) of the verification problem to see what the difference in performance is. This can indicate how much benefit is obtained by conditioning on the neural network graph as the input representation and the associated forward-backward message passing schedule.
- How accurate is the learned heuristic in imitating strong branching? Is it necessary to get high accuracy on the imitation task to achieve an improvement in the final solve task?
- As a baseline it would be good to include the results for branch-and-bound using strong branching. Even if this is much slower, it would still help to know how much slower.
- I’m surprised that the reduction in the number of branches closely follows the reduction in the running time. This seems to suggest that the overhead of running graph neural network inference within branch-and-bound is negligible. Is this the case? If so, why -- is it because the LP solve time is much higher than the graph net inference time?


**Experience Assessment:**

I have read many papers in this area.

**Review Assessment: Checking Correctness Of Derivations And Theory:**

I assessed the sensibility of the derivations and theory.

**Review Assessment: Checking Correctness Of Experiments:**

I carefully checked the experiments.

**Review Assessment: Thoroughness In Paper Reading:**

I read the paper at least twice and used my best judgement in assessing the paper.

---

> ### Author Response · Authors · 2019-11-14
> **Thank you for review, comments and questions.**
>
> We have included results on MNIST dataset in the appendix of the updated paper.
>
> In regards to additional comments:
>
> 1.	Geometric means are now reported in the appendix.
>
> 2.	There are three main problems with Gasse’s approach when apply it on Neural Networks. Firstly, due to its generality, Gasse’s bipartite graph does not effectively capture the structure of a Neural Network. Variables corresponding to input nodes, pre-activation nodes, post-activation nodes and output nodes are all treated the same and updated by the same update function, despite that each type of variables are considerably different from others. This would result in reduced expressiveness and model capacity of Gasse’s approach and increased difficulty in learning effective parameters for update functions. The same is true for all constraints and their associated update functions.
> 	Secondly, during each pass, embedding vectors for all variables (constraints) are updated at the same time for bipartite graphs. We have mentioned in the GNN section that the amount of relaxation introduced at each ambiguous node is largely affected by information on previous layers. Bipartite graphs cannot express the forward nature of impacts.
> 	Lastly, as a branching tool developed for mixed-integer linear programs, Gasse’s approach relies heavily on features obtained through solving LPs, especially for constraints. This limited its flexibility, given that LP problems of neural networks are generally large and expensive to solve. Our GNN framework still outperforms the Baseline when all LP features are removed. To demonstrate this, we trained a new GNN with LP features removed and tested the new GNN on 260 randomly selected verification properties on the Base model. Among the selected properties, 140 are categorised as easy, 70 as medium and 50 as hard. We denote the model trained on all features as GNN and the newly trained model as GNN-R (we use R to indicate reduced features).
>
> 				        Easy		Med		Hard
> GNN : time (s)		268.60		724.88		1025.83
> GNN-R : time (s)		348.48		898.01		1340.56
> BABSR : time (s)		429.59		1622.67	        2466.71
>
> GNN : branches		319.39		529.07		772.67
> GNN-R : branches        441.04		720.96		967.80
> BABSR: branches		641.30		1504.37	      1931.10
>
> Although removing primal and dual information deteriorates the GNN performance, it is clear that GNN-R performs better than the baseline heuristic BABSR.
>
> 3.	We now report the accuracy of the learned heuristic in imitating strong branching in the Appendix of the updated paper. We mention that since there are several branching choices that give similar performances at each subdomain, we considered all branching choices that have m_v (defined in (8)) above 0.9 as correct decisions. Under this assumption, our trained GNN achieves 85.8% accuracy on the training dataset and 83.1% accuracy on the validation dataset. We believe it is important to obtain high accuracy on imitation task for GNN to work.
>
> 4.	We do not include strong branching as one of our baselines because, given the large amount of ambiguous nodes and the LP cost, it is computationally infeasible to use strong branching for branch-and-bound verification. It is also the reason that we proposed a new strategy for generating training data cheaply and inclusively.
>
> 5.	Yes, LP solving time is the main bottleneck for branch-and-bound based verification methods. Although both GNN evaluation time and LP solving time increase with the size of network, LP solving time grows at a significantly faster speed. For instance, in CIFAR experiments, GNN requires on average 0.02, 0.03, 0.08 seconds to make a branching decision on Base, Wide and Deep model respectively but the corresponding one LP solving time on average are roughly 1.1, 4.9, 9.6 seconds. GNN evaluation is almost negligible for large neural networks when compared to LP solving time.

---

### Official Review · AnonReviewer2 · 2019-10-24
**Official Blind Review #2**

**Rating:** 6

**Review:**

This paper proposes to use graph neural networks (GNNs) to replace the
splitting heuristic in branch and bound (BaB) based neural network verification
algorithms. The paper follows the general BaB framework by Bunel et al., but
considers only splitting ReLU neurons, not input domains. The GNN is built by
replacing each neuron in network to be verified as a vertex, and the
connections between neurons as edges. Each vertex has a feature vector combining
information like pre-activation bounds and primal/dual LP solutions.  A
specialized GNN training procedure is developed to exploit the structure of the
problem, and the weights of GNN are updated in a forward and backward manner.

Overall the paper proposes a novel idea of using GNN for accelerating
verification and it is demonstrated to be effective on one MNIST network as
well as its wider and deeper variants.  I feel the main weakness is that the
empirical evidence provided are not thorough and sufficient (only 1 base model
on 1 dataset). Since this paper is 10-page, I evaluate it at a higher standard
and expect more convincing empirical results.

Questions and suggestions for improvements:

1. How much time does it take to generate training examples? It seems to me
that it is a very costly process because obtaining the relative improvement (8)
of splitting at each node can be quite expensive - basically, we need to split
almost every ambiguous neuron to get their improvement values, and in normal
BaB we only need to split one each time. The paper mentioned it "minimum 5%
coverage per layer" but does not provide more details.

2. Also how much time does it take for training the GNN? It seems the GNN has
many vertices - the same number as the number of neurons in a network, which
can be quite large.  If the dataset generation and training time are much
longer comparing to the BaB time, the usefulness of the proposed method can be
limited especially it does not necessarily generalize to foreign networks
(networks with significantly different structure, or trained using different
methods).

3, An ablation study for the fail-safe strategy is needed. Without the
fail-safe strategy, is the GNN learned split better than other strong
heuristics? If the fail-safe strategy is too strong, the improvement we see can
probably come from the fail-safe strategy mostly, and GNN might not do too much
useful things. This is an important study that should be part of this paper.

4. It seems all networks in this work are trained using a single training
method, Wong & Kolter, 2018.  Does the split heuristic learned by GNN works for
networks trained using a different training strategy?  For example, interval
bound propagation (IBP) based methods [1][2] which achieve the state-of-the-art
results.  Also, adversarial training with L1 regularization is also verifiable,
as demonstrated in [3]. Running a few pretrained models by these methods should
be an easy experiment to add.

5. There have been a few strong baselines in this field that the authors do not
discuss and compare against, including [4][5][6]. They solve similar problems
as in this paper and also provide promising results. At least, the authors should
discuss them in related works, and it is strongly encouraged to add at least one
of them as a stronger baseline.

6. This paper claims that Neurify is theoretically incorrect (in Appendix D.2,
page 17). I am quite surprised and not sure if this claim is true. I am not
aware of any firm evidence that Neurify is theoretically incorrect.  It is
better to communicate with the authors of Wang et al., 2018 and make sure this
paper is making a correct claim.

Given that the idea proposed by this paper is novel and interesting, I tend to
accept this paper *under the condition* that the authors can conduct an
ablation study of the fail-safe strategy, provide generalization results on
models trained using different robust training strategies, and provide results
on at least one more dataset (like ACAS Xu, or CIFAR). Adding at least one more
baseline is also strongly encouraged.

** After author discussion, I have increased my score based on the new results
provided. The authors should make sure to include the ablation study results, and
a detailed discussion on training data generation time and training time in the
final version of the paper.



[1] Sven Gowal, Krishnamurthy Dvijotham, Robert Stanforth, Rudy Bunel, Chongli Qin, Jonathan Uesato, Timothy Mann, and Pushmeet Kohli. "On the effectiveness of interval bound propagation for training verifiably robust models." arXiv preprint arXiv:1810.12715 (2018).

[2] Huan Zhang, Hongge Chen, Chaowei Xiao, Bo Li, Duane Boning, and Cho-Jui Hsieh, "Towards Stable and Efficient Training of Verifiably Robust Neural Networks" (https://arxiv.org/abs/1906.06316)

[3] Xiao, K. Y., Tjeng, V., Shafiullah, N. M., & Madry, A. (2018). Training for faster adversarial robustness verification via inducing relu stability. arXiv preprint arXiv:1809.03008.

[4] Katz, Guy, et al. "The marabou framework for verification and analysis of deep neural networks." International Conference on Computer Aided Verification. Springer, Cham, 2019.

[5] Singh, G., Gehr, T., Püschel, M., & Vechev, M. (2018). Boosting Robustness Certification of Neural Networks.

[6] Anderson, G., Pailoor, S., Dillig, I., & Chaudhuri, S. (2019, June). Optimization and abstraction: a synergistic approach for analyzing neural network robustness. In Proceedings of the 40th ACM SIGPLAN Conference on Programming Language Design and Implementation (pp. 731-744). ACM.


**Experience Assessment:**

I have published in this field for several years.

**Review Assessment: Checking Correctness Of Derivations And Theory:**

I carefully checked the derivations and theory.

**Review Assessment: Checking Correctness Of Experiments:**

I carefully checked the experiments.

**Review Assessment: Thoroughness In Paper Reading:**

I read the paper thoroughly.

---

> ### Author Response · Authors · 2019-11-14
> **Thank you for your review, questions and suggestions.**
>
> We have demonstrated the effectiveness of our GNN framework on the challenging CIFAR dataset over three different network architectures. Unlike most other adversarial papers, which use fixed epsilon values, we tackled verification properties with difficult epsilon values, specifically selected through binary search.
>
> With regards to questions and suggestions for improvements, we reply to each comment following the same numbering used by the reviewer.
> 1.	We used the base model, which has 3172 hidden nodes, to generate training data. For a typical hard epsilon value, each sub-domain generally contains 1300 ambiguous ReLU nodes. Among them, approximately 140 ReLU nodes are chosen for strong branching heuristics, which leads to roughly 200 seconds for generating a training sample. We point out that the total amount of time required for generating a training sample equals the 2*(per LP solve time)*(number of ambiguous ReLU nodes chosen). Although both the second and the third terms increase with the size of the model used for generating training dataset, the vertical transferability of our GNN enables us to efficiently generate training dataset by working with a small substitute of the model we are interested in.
> 2.	We have divided the nodes on GNN into input nodes, activation nodes and output nodes. Since the nodes of the same type share the same set of parameters, the total number of parameters of GNN that need to be learnt is small and does not depend on the input Neural Network size. Our proposed GNN is fast to train. In our experiments, each training epoch took less than 400 seconds and the GNN converges within 60 epochs.
> 3.	In all our experiments, we have compared against BABSR, which employs only the fail-safe heuristic for branching. In other words, removing the GNN and using only the fail-safe heuristic is equivalent to BABSR. The fact that GNN significantly outperforms BABSR demonstrates that GNN is doing most of the job. Based on your suggestion, we collected the following statistics to show the effectiveness of our GNN. For each experiment, we computed the ratio of times GNN is used and the fail-safe heuristics is used respectively on all verification properties.
>       		%GNN decision is used	%Fail-safe heuristics is used
>       BASE 		0.9342				0.0658
>       WIDE		0.9497				0.0503
>       DEEP		        0.9638				0.0362
> 4.	 We emphasis that for adversarially and robustly trained networks, the difficulty of a property relies on the value of epsilon. In terms of adversarially trained networks, we followed the training method in [1] and trained a model with epsilon 8/255 using the small CIFAR architecture provided in [2]. Regarding robustly trained models, we downloaded the trained ConvMed CIFAR model with epsilon = 8/255, provided in [3]. For both models, we selected 1000 verification properties with epsilon=8/255 at random. For almost all of them, no branching was required to either prove or disprove a properties as the initial lower bound obtained via linear programming by Gurobi was sufficient.
> We need challenging verification properties to evaluate different branching strategies. We hence point out that one of our contributions is that we have generated difficult properties by searching over difficult epsilon values via expensive binary search. We hope that our curated data set would prove useful in comparing other verification algorithms.
> 5.	Thank you for suggesting the related works, which we have now cited in the updated version. In order to strengthen the baseline, we compared our approach to a new MIP based algorithm proposed in [4] and tested it on 100 randomly selected properties from the Base experiment. Compared to MIPplanet, which requires 1732.18 seconds on average, [4] requires 2736.60 seconds. We mention that the main difference between MIPplanet and [4] is the intermediate bound computation, which is complementary to our focus. If better intermediate bounds are shown to help verification, we can still use our approach to get better branching decisions corresponding to those bounds.
> 6.	Regarding Neurify, the previous code release for the paper did not explicitly perform verification. The updated version does indeed perform verification. We have added a comparison to Neurify in the appendix, and corrected our text.
> We have included experiments on MNIST in the appendix of the new version of the paper.
> [1] Madry, Aleksander, et al. "Towards deep learning models resistant to adversarial attacks."  ICLR 2018.
> [2] Wong, Eric, et al. "Scaling provable adversarial defenses." NeurIPS. 2018.
> [3] Mirman, Matthew, Timon Gehr, and Martin Vechev. "Differentiable abstract interpretation for provably robust neural networks." ICML. 2018.
> [4] Tjeng, Vincent, Xiao Kai, and Tedrake Russ. "Evaluating robustness of neural networks with mixed integer programming." ICLR. 2019.

---

### Official Review · AnonReviewer4 · 2019-10-27
**Official Blind Review #4**

**Rating:** 8

**Review:**

Summary:
This paper deals with complete formal verification of Neural Network, based on the Branch and Bound framework. The authors focus on branching strategies, which have been shown to be a critical design decision in order to obtain good performance. The tactic employed here is to learn a Graph Neural Network (which allows to transfer the heuristic from small networks to large networks), using supervised training to imitate strong branching. The authors also discuss fallback mechanism to prevent bad failures case, as well as an online fine-tuning strategy that provide better performance.
Experiments are performed on the CIFAR dataset and show convincing improvements compared to the baselines.

Comments:
* "This allows us to harness both the effectiveness of strong branching strategies and the efficiency of GPU computing power". Most other hand crafted heuristics also benefit from GPU computing, as they are based on gradients, or on the K&W dual, which all have GPU implementations.
* The description of the Nodes indicates that all hidden activation have a representative node in the GNN. Does it make sense to have it for non-ambiguous hidden activations?
* "Since intermediate lower and upper bounds of a node xˆi[j] are completely decided by the layers prior to it" -> That's not necessarily true depending on the Relaxation used. In the context of the full LP relaxation of Ehlers and branching on the ReLUs, constraint on following nodes can have an impact on earlier bounds. The authors make the same point later in the paragraph, so it's just a matter of being precise in the writing.
* "underlying data distribution, features and bounding methods are assumed to be same when the trained model is applied to different networks" -> This is a very reasonable assumption to make. Is there some intuition on which features are the most important? Given the features chosen, a strong relaxation needs to be used to obtain all the required features. Do the authors have any insights or experiments on how looser relaxations, which would lead to less feature available would fare?
* With regards to the improvement measure (8), I'm slightly confused by the definition. It essentially measures independently and averages the improvement for each of the subdomain resulting of the split. In this case, if we go from one subdomain with a lower bound of -5, to a pair of subdomain with respective lower bounds (0, -5). (essentially we have split across a useless dimension), this metric will grant a certain amount of improvements, while the global lower bounds held by the BaB process will not have changed. Did the authors give a try to other metrics?
* I'm happy to see some discussion of the failures case of following a learned policy, leading to a series of bad decisions, which in my experience is a real problem. Am I correct in understanding the explanation that after a split is done, if it provides poor improvement, the split is undone and a back-up heuristic is applied? Or is it just that for the resulting subdomains of the low improvement split, the back-up heuristic is used?

* I'm wondering if some hand crafted heuristics could be learned by the model? As in, is the model expressible enough that it could encode the heuristics of Bunel, Royo or Wang? This would be an interesting analysis and show that following the learning approach is essentially a "free win". From what I can see, it wouldn't be able to as it is missing some information (the GNN doesn't have access to the bias of the network for example?), but I might be wrong.

* For the upper bound computations, "For the output upper bound, we compute it by directly evaluating the network value at the input provided by the LP solution". Is there some reference on how effective of a scheme that is, compared to more expected things like adversarial attacks?

* I know that they are not directly comparable but Gurobi provides the information about the numbers of branches that it performed internally. This would have been beneficial to obtain for the results of Table 1 and 2

* Am I correct in assuming that MIPplanet is the same method as in Bunel et al., where all intermediates bounds are computed with the method of Ehlers et al.? Given that solving LP on large networks can be quite slow, is this method penalized by using tight but very expensive bounds? Would a MIP with bounds based on the linear relaxation of (1b) be faster and provide a stronger baseline?

* One aspect that is missing from this paper is the discussion of the cost of generation of the training dataset, and of the training of the GNN? How many properties do you need to have to verify for it to make sense to learn a heuristic rather than just using a handcrafted one?
There might also be some more interest if the network was shown to generalize to other settings. We can already observe that there is at least some transfer between architectures and across "hardness of problems", but it would be great to see if it generalizes further (learn a GNN on MNIST, use it to verify CIFAR?)

Opinion:
The paper is quite interesting and outperform its baseline by a significant amount. I have some question about whether the MIP baseline is the best one but even if it could have been improved, I still think there is interest in methods that are more specialized and go beyond trusting a MIP solver.

**Experience Assessment:**

I have published in this field for several years.

**Review Assessment: Checking Correctness Of Derivations And Theory:**

I carefully checked the derivations and theory.

**Review Assessment: Checking Correctness Of Experiments:**

I carefully checked the experiments.

**Review Assessment: Thoroughness In Paper Reading:**

I read the paper thoroughly.

---

> ### Author Response · Authors · 2019-11-14
> **Thank you for review, comments and questions**
>
> 1,3	Regarding harnessing GPU computing power and the relationship between intermediate bounds and prior layers, we thank the reviewer for the clarification. We have incorporated these changes in the new version of the paper.
>
> 2.	With regards to non-ambiguous hidden activations, we found it useful to include the ones with lower bounds above zero as they allow complete information to pass by outputting a value equal to the input value. However, it is true that hidden nodes with upper bounds below zero can be removed from the GNN, as they act as complete blocking gates. In our experiments, we fix the embedding vectors for blocking ReLU nodes to zero vectors. In the main paper, we give general descriptions of non-ambiguous nodes for the sake of simplicity and conciseness. Detailed treatments of these nodes are covered in the appendix.
>
> 4.	In terms of important features, we mention that we used two types of features. The first type (including intermediates bounds, network weights and biases) can be collected at negligible costs. The other type is LP features (primal and dual values) that are acquired by solving a strong LP relaxation, which are expensive to compute but potentially highly informative. To evaluate their effect, we trained a new GNN with LP features removed and tested the new GNN on 260 randomly selected verification properties on the Base model. Note that we use a random subset of test properties only due to time constraints.
> Among the selected properties, 140 are categorised as easy, 70 as medium and 50 as hard. We denote the model trained on all features as GNN and the newly trained model as GNN-R (we use R to indicate reduced features).
>
> 				          Easy		Med		Hard
> GNN : time (s)		268.60		724.88		1025.83
> GNN-R : time (s)		348.48		898.01		1340.56
> BABSR : time (s)		429.59		1622.67	         2466.71
>
> GNN : branches		319.39		529.07		772.67
> GNN-R : branches	441.04		720.96		967.80
> BABSR: branches		641.30		1504.37	       1931.10
>
> Removing primal and dual information deteriorates the GNN performance, but GNN-R still outperforms the baseline heuristic BABSR. We believe cheap features are the most important.  Depending on the cost of LP, potential users can either remove expensive LP features or train a GNN with a smaller architecture.
>
> 5.	In terms of improvement measure (8), for the given example, although the global lower bound is not increased, the subdomain with lower bound 0 will be pruned away after the split, which is still a valid improvement in terms of narrowing down the problem domain. We have tried various metrics, including picking the minimum of the two subdomain lower bounds and the maximum of the two lower bounds. Among these metrics, metric defined by (8) performs the best.
>
> 6. 	Regarding fail-safe heuristics, the reviewer’s understanding is correct: the heuristic is used for the parent domain of the low improvement split. Fail-safe strategy is important for the success of our GNN framework. Without a fail-safe strategy, we have observed cases that GNN timed out on properties easily solved by the heuristic for the reasons mentioned in the Fail-safe Strategy paragraph in the paper. However, we emphasise that GNN outperforms the baseline heuristic significantly once fail-safe strategy is employed. Among all verification properties tested, back-up heuristic is used roughly 5% on average.
>
> 7. 	In terms of learning hand-crafted heuristics, we point out that GNN has access to all the weights and biases of the network. The biases form part of the node features. The heuristics of Bunel, Royo and Wang use either a forward pass and/or a backward pass to make a branching decision. Since our GNN has a forward and backward passing updating schedule and we have included all features used in those hand-designed heuristics, our GNN is expressible enough to encode those heuristics.
>
> 8.	With regards to upper bound computations, during our experiments, we have also tried finding a counter-example via adversarial attacks and random sampling. We found our current implementation of computing upper bound gives the tightest upper bounds and hence suits our datasets the best.
>
> 9.	We have recorded branch number outputted by Gurobi for a subset of properties for each model and reported the number in the appendix. We found that Gurobi branch number is not positively related to the solving time. We suspect Gurobi performs cutting before branching, so time spent on branching varies between properties, leading to inconsistent branch number and solve time.

---

> > ### Author Response · Authors · 2019-11-14
> > **Reply to the rest of comments**
> >
> > 10. 	The MIPplanet is similar to the method in Bunel et al. The key difference is that, on each subdomain, intermediate bounds are computed with the linear relaxation (1b) to form the final mixed integer program instead of using an LP relaxation for the intermediate bounds. As the reviewer suggests, this is indeed a stronger baseline than the original MIPplanet as the time required for each bound computation reduces significantly.
> >
> > 11. 	Cost of generating training dataset is now reported in the Appendix of the updated paper. We used base model, which has 3172 hidden nodes, to generate training data. For a typical hard epsilon value, each sub-domain generally contains 1300 ambiguous ReLU nodes. Among them, approximately 140 ReLU nodes are chosen for strong branching heuristics, which leads to roughly 200 seconds for generating a training sample. We point out that the total amount of time required for generating a training sample equals the 2*(per LP solve time)*(number of ambiguous ReLU nodes chosen). Although both the second and the third terms increase with the size of the model used for generating training dataset, the vertical transferability of our GNN enables us to efficiently generate training dataset by working with a small substitute of the model we are interested in.
> > 	There are various factors that come into a decision of when to use a learnt model than a heuristic. Learnt GNN generally performs better than a hand-designed heuristic. Although generating training dataset could be expensive, potential users can train a GNN on a smaller network architecture to speed up the process. There is indeed a tradeoff between time and accuracy. The exact decision of choice should depend on the problem at hand and it is not possible to set a hard decision threshold on number of verification properties.
> >
> >
> > General Comments:
> >
> > 1.	In regarding to the general comment of testing the generalization to other settings, we tested CIFAR trained GNN on MNIST verification properties. In detail, we have tested the GNN on 20 randomly picked verification properties of MNIST Base model. We found that BABSR outperforms CIFAR trained GNN on all properties, so the CIFAR trained GNN model does not transfer to MNIST dataset. This is expected as MNIST and CIFAR images differ significantly from each other.
> >
> > 2.	In terms of whether the MIP baseline can be improved, we have also compared against a MIP solver [1] from ICLR 2019. We tested it on 100 randomly selected properties from the CIFAR Base experiment. We found that [1] is in fact slower than our MIPplanet baseline (2736.60 seconds for [1] vs. 1732.18 for MIPplanet). We mention that the main difference between MIPplanet and [1] is the intermediate bound computation, which is complementary to our focus. If better intermediate bounds are shown to help verification, we can still use our approach to get better branching decisions corresponding to those bounds.
> >
> > [1] Tjeng, Vincent, Kai Xiao, and Russ Tedrake. "Evaluating robustness of neural networks with mixed integer programming." International Conference on Learning Representations. 2019..

---

> > > ### Comment · AnonReviewer4 · 2019-11-14
> > > **Thank you for your reply**
> > >
> > > Thank you for your detailed replies to my question. I will update my score positively.
> > >
> > > Please include at least a mention of the non-generalization of networks between datasets somewhere in the paper, I could not find it in the updated version.
> > > It would be also good if you had a lengthier discussion/comparison with the work of Anderson et al, as they also perform some sort of learning to branch.

---

> > > > ### Author Response · Authors · 2019-11-14
> > > > **Thank you for your comments.**
> > > >
> > > > Thank you very much for your additional comments. We have updated the paper accordingly.

---

### Public Comment · ~Gagandeep_Singh1 · 2019-10-22
**Question about generalization and missing comparison with state-of-the-art**

This is an interesting work and appears like a step in making a dedicated solver for neural network verification. Though, I like the idea for your new approach, I have a few concerns. It appears that the test set is not very different from the training set. All networks used in the experiments are trained with the same training procedure. This can lead to overfitting where the verification procedure gives worse results on networks trained with normal training or with another certified training method (e.g. Wong’s verification method gives close to complete results on the network trained with Wong's method and not on other methods). How would the results would look like on publicly available networks from [1,2,3].

Also I am not sure if the baselines you compare against are state-of-the-art MILP methods for neural network verification. The networks used here are much smaller compared to [4,5,6] that also use MILP solving. Can you compare against the following works?

References:
[1] Towards Deep Learning Models Resistant to Adversarial Attacks, ICLR 2018.
[2] Differentiable abstract interpretation for provably robust neural networks. ICML 2018.
[3] https://github.com/eth-sri/eran
[4] Evaluating robustness of Neural networks with Mixed Integer Linear Programming. ICLR 2019.
[5] Boosting Robustness Certification of Neural Networks. ICLR 2019.
[6] On the Effectiveness of Interval Bound Propagation for Training Verifiably Robust Models. Arxiv 2018.

---

> ### Author Response · Authors · 2019-11-05
> **Thank you for your interest and comments.**
>
> (1) Our experimental setup does not represent a case of overfitting, rather a practically useful scenario where we need to verify several properties for the same network. Indeed, for the Base experiment, the training and testing properties are obtained from a different set of CIFAR images. Furthermore, learning from a small network generalises to wider/deeper networks, thereby making our approach widely applicable. For example, in order to verify networks trained using a different approach (e.g. adversarial training or abstract interpretation) we may require a different GNN, but this can be obtained in practice by efficiently generating a training data set using a small architecture trained with the same approach.
>
> (2) We emphasis that the difficulty of a property not only relies on the size of the network, but also the value of epsilon. For [1], we trained the model with epsilon 8/255 and used the small CIFAR architecture provided in (Scaling provable adversarial defenses https://arxiv.org/abs/1805.12514). For [2], we downloaded the trained ConvMed model and used epsilon = 8/255. For both models we selected 1000 testing properties at random. For almost all of them, no branching was required to either prove or disprove a properties as the initial lower bound obtained via linear programming by Gurobi was sufficient. In contrast, we have generated difficult properties by searching over difficult epsilon values via binary search. We hope that our curated data set would prove useful in comparing other verification algorithms.
>
> (3) Thank you for suggesting other baselines. We used [4] to solve 100 randomly selected properties from the Base experiment. Compared to MIPplanet, which requires 1732.18 seconds on average, [4] requires 2736.60 seconds. The large runtime of [4] may be due to the fact that our epsilon values are explicitly chosen to explore difficult verification cases. Note that the main difference between MIPplanet and [4] is the intermediate bound computation, which is complementary to our focus. If better intermediate bounds are shown to help verification, we can still use our approach to get better branching decisions corresponding to those bounds.

---

### Decision · Program_Chairs · 2019-12-19

**Decision:**

Accept (Talk)

**Comment:**

The authors develop a strategy to learn branching strategies for branch-and-bound based neural network verification algorithms, based on GNNs that imitate strong branching. This allows the authors to obtain significant speedups in branch and bound based neural network verification algorithms relative to strong baselines considered in prior work.

The reviewers were in consensus and the quality of the paper and minor concerns raised in the initial reviews were adequately addressed in the rebuttal phase.

Therefore, I strongly recommend acceptance.